# Patient-Reported Outcomes (PROs) and Patient Experiences in Fertility Preservation: A Systematic Review of the Literature on Adolescents and Young Adults (AYAs) with Cancer

**DOI:** 10.3390/cancers15245828

**Published:** 2023-12-13

**Authors:** Nicole F. Klijn, Moniek M. ter Kuile, Elisabeth E. L. O. Lashley

**Affiliations:** Department of Gynecology and Obstetrics, Leiden University Medical Center, Albinusdreef 2, 2333 ZA Leiden, The Netherlands

**Keywords:** fertility preservation, PROs, patient experiences, AYA, oncofertility

## Abstract

**Simple Summary:**

With better survival rates for patients diagnosed with cancer, more attention has been put on future risks like fertility decline due to gonadotoxic treatment. This review focusses on patient-reported outcomes (PROs) and patient experiences among adolescent and young adults (AYAs) diagnosed with cancer. An extended search showed that health care providers need to acknowledge the importance of future fertility and discuss with every AYA the potential of fertility decline. AYAs often requested a referral to a fertility specialist to be informed about fertility preservation (FP) options. They also commonly asked for more patient-specific (written) information about FP options. A clear FP pathway can prevent a delay in receiving a referral to a fertility specialist to discuss FP options and initiating FP treatment. This patient-centered approach will optimize FP experiences and establish a process to achieve long-term follow up after FP treatment.

**Abstract:**

With better survival rates for patients diagnosed with cancer, more attention has been focused on future risks, like fertility decline due to gonadotoxic treatment. In this regard, the emphasis during counselling regarding possible preservation options is often on the treatment itself, meaning that the medical and emotional needs of patients regarding counselling, treatment, and future fertility are often overlooked. This review focuses on patient-reported outcomes (PROs) and patient experiences regarding fertility preservation (FP)—among adolescents and young adults (AYAs) with cancer. A systematic review of the literature, with a systematic search of online databases, was performed, resulting in 61 selected articles. A quality assessment was performed by a mixed methods appraisal tool (MMAT). Based on this search, three important topics emerged: initiating discussion about the risk of fertility decline, acknowledging the importance of future fertility, and recognizing the need for more verbal and written patient-specific information. In addition, patients value follow-up care and the opportunity to rediscuss FP and their concerns about future fertility and use of stored material. A clear FP healthcare pathway can prevent delays in receiving a referral to a fertility specialist to discuss FP options and initiating FP treatment. This patient-centered approach will optimize FP experiences and help to establish a process to achieve long-term follow up after FP treatment.

## 1. Introduction

With better survival rates for patients diagnosed with cancer in recent decades, the potential future risks of cancer treatments are now receiving more attention. One clearly defined risk is a potential decline in fertility, due to the gonadotoxic effects of chemotherapeutic agents.

In 2006, the term oncofertility was introduced by Dr. Teresa Woodruff to highlight the importance of discussing future fertility and the future reproductive health of young women before or during oncological treatments [1]. The resulting oncofertility consortium aims to provide fertility preservation options in a cancer-therapy setting [2]. With oncofertility embedded in oncological treatment policies, in combination with collaboration with multidisciplinary teams when discussing adolescent and young adult (AYA) patients diagnosed with cancer, an increasing number of patients are now being informed about the risk of infertility and their fertility preservation (FP) options. This information is essential prior to starting a gonadotoxic treatment [3,4,5,6]. FP counseling has been shown to result in less decisional regret and better quality of life [3,7], even if FP is not an option for an individual patient [8].

However, FP counseling is complex. Decisions must be made rapidly before starting a gonadotoxic treatment; under these circumstances, patients are emotionally stressed due to a recent cancer diagnosis, and their decisions are considered to be “eternally binding”. In reality, the emphasis during counseling is often on the treatment itself, with little time left to discuss the further medical and emotional needs of patients regarding counseling, treatment, and future fertility [9].

To transition from an illness-oriented to a more patient-centered approach, value-based healthcare (VBHC) was introduced as a new method for clinical decision making. In this method, all values, especially when they are complex and sometimes conflicting, are ranked [10]. These values include traditional health outcomes, such as pregnancy or complication rates, and values based on outcomes and experiences which are important to the patient.

In this review, we aim to provide an overview of patient-reported outcomes (PROs) and patient experiences regarding counseling, treatment, and future fertility in FP. This review specifically focuses on the experiences and needs of AYA patients, generally defined as cancer patients aged 15–39 years who need to start an oncological treatment. We narrowed the search to this specific group because we are currently developing and optimizing a healthcare pathway specifically for this patient group. The PROs and patient experiences defined by this patient group will be used to create patient-reported outcome measures (PROMs) and patient-reported experience measures (PREMs). These PROMs and PREMs can be incorporated in this healthcare pathway and will add value to our care.

## 2. Materials and Methods

We conducted a systematic review following the PRISMA guidelines [11]. This systematic review was registered and accepted for inclusion in the PROSPERO international prospective register of systematic reviews (ID CRD42023434721).

### 2.1. Search Strategy and Study Selection

A systematic search was conducted on 10 August 2023 of the PubMed, Embase, and Web of Science electronic databases. We used the following free text and MeSH terms: fertility preservation, sperm, oocyt*, testic*, embryo*, cryopreservation, oncofertility, assisted reproduction, neoplasms, cancer, patient-reported outcome, patient-reported experience, qualitative research, surveys, and questionnaires. The full electronic search strategy for PubMed is shown in Appendix A. The reference lists of the identified articles were manually searched for additional relevant references. A re-run of the search was performed on 27 October 2023, prior to the final analysis.

### 2.2. Eligibility Criteria

The inclusion criteria were original research articles in English, addressing PROs and patient experiences of AYAs regarding FP counseling, treatment, and future fertility. We included studies with AYAs, defined as patients between 15 and 39 years of age at diagnosis, or studies where >75% of the included patients were in this age range. The exclusion criteria were studies that focused on awareness of FP options or referral pathways, as well as articles about younger patients (defined as <15 years), articles about children and their parents, and articles describing research in which the main group of patients was older than 40 years of age.

The literature search was performed by a librarian and two researchers (N.K. and E.L). The results were exported to the EndNote citation manager. Duplicates were removed. Screening of the articles was performed by two researchers (N.K. and E.L.); this consisted of two stages. In the first stage, titles and abstracts were screened. In the second stage, manuscripts were reviewed in full. Any disagreements in terms of selecting studies or assessing their quality (see further) were resolved by consensus. If no agreement was obtained, the opinion of a third researcher (M.K.) was sought.

### 2.3. Data Analysis

Studies relevant for PROs and patient experiences that contained relevant information about female patients only, male patients only, or a combination of female and male patients were selected. The data extracted from each eligible study were recorded in a standardized extraction table including study design, country, publication year, study period, population characteristics, year of FP counselling, inclusion criteria, and relevant findings. These data were synthesized using narrative descriptions.

### 2.4. Quality Assessment

The quality of the selected studies was assessed using the Mixed Methods Appraisal Tool (MMAT) [12]. This is a widely used instrument for quality appraisals of quantitative, qualitative, and mixed methods studies. It can be used to assess the methodological quality of five categories of studies: qualitative research, randomized controlled trials, non-randomized trials, quantitative descriptive studies, and mixed methods studies. Each selected study was rated according to the MMAT guidelines. A qualitative assessment of each study was performed and the results were discussed by two authors (N.K and M.K). If no agreement was reached, the opinion of a third observer (E.L.) was sought to gain consensus.

## 3. Results

Our systematic database search retrieved a total of 2181 articles of which 1184 were unique and retained for first stage screening. After the first stage screening, 147 articles were selected for stage-two assessment. A re-run of the search was performed before finalization of the review. This resulted in 2 extra articles for full review. After the second stage screening, 61 articles that met all inclusion criteria were selected. These studies contained information regarding PROs and patient experiences; 36 were about female patients, 12 were about male patients, and 13 were about both female and male patients. Details of the study selection process are shown in the PRISMA Flow Diagram in Figure 1.

### 3.1. Study Characteristics

All included studies were published between 2002 and 2023. Thirty-one qualitative studies [8,9,13,14,15,16,17,18,19,20,21,22,23,24,25,26,27,28,29,30,31,32,33,34,35,36,37,38,39,40,41], 26 quantitative studies [42,43,44,45,46,47,48,49,50,51,52,53,54,55,56,57,58,59,60,61,62,63,64,65,66,67], and 4 mixed method studies [68,69,70,71] were included. The studies had different designs, i.e., retrospective cohort, cross-sectional, and prospective cohort. In retrospective studies, patients were sampled, and often, information was collected through interviews about FP counselling and treatment. In cross-sectional studies, the outcomes and experiences of all patients were measured at or around the moment of FP counselling/treatment. In the prospective studies, patients were followed over time and data were collected at different time points for the same individuals. Most of the included studies were conducted in Western Europe (*N* = 26), followed by North America (*N* = 20), Asia (6), Australia (6), and Africa (1). In two studies, patients were recruited on different continents. In 47 studies, heterogeneous samples of different cancer diagnoses were included. Fourteen studies included patients that had the same diagnosis, most often breast cancer (79%). Two studies withheld detailed information about cancer diagnoses. The mean age of patients at diagnosis ranged from 17 to 35 years. The time between diagnosis and examination varied between a couple of weeks and up to more than 25 years. Descriptions of the key characteristics of all included studies are provided in Table 1.

### 3.2. Risk of Bias

As the studies included in this research differed in their methodologies, we used the MMAT for our qualitative assessments. As shown, we included studies with qualitative, quantitative, and mixed methods designs. Sample sizes of interview studies varied from low, with the question of whether saturation for conclusions has been reached, to reasonable and good. In quantitative studies, response rates varied between 22% and 86%. Non-response bias can have an effect on the documented results. Details of our methodological analysis are described in Appendix A.

### 3.3. Narrative Synthesis

The data extracted from the included studies were synthesized using narrative descriptions. Table 2 summarizes the relevant PROs and patient experiences among AYAs in FP.

### 3.4. Starting a Conversation about Potential Fertility Decline and Referral for Fertility Preservation Counselling

#### 3.4.1. Acknowledging the Importance of Future Fertility

Studies involving patients diagnosed with a disease whose treatment can be gonadotoxic indicated that talking about potential fertility decline is important for patients. Multiple studies reported that maintaining fertility has significant meaning for both male and female patients [25,32,53,62,64]. Latif et al. noted that male AYAs consider cancer-related infertility an important issue, as for most, fatherhood is of immense significance [32]. Schover et al. reported this specifically in men who were childless at the time of their cancer diagnosis [62]. Studies in female AYAs, on the other hand, reported that women experience fertility as an essential element of their femininity [20,25]. Kirkman et al. noted that women appreciate having the importance of FP and future childbearing recognized. If fertility concerns are not well managed, patients feel troubled [24]. Others also indicated that sometimes health care providers made treatment decisions focused on survival or extending life on behalf of the patient. Patients in these cases reported that they felt they had no choice in relation to health professional decision making. This prevented them from taking action to preserve their fertility [27,39]. Women explained that the focus of health care providers was on treating the cancer and getting practical things organized, rather than the impact of the diagnosis and treatment on the woman herself [17]. In addition, multiple studies described that female and male patients initially did not consider parenthood to be important. Going through cancer and threatened fertility had altered their perspective of how important it is to be a future parent [15,19,30,35,41]. For all these reasons, it is important to give patients with cancer the chance to think about their future parenthood [20,31]. Physicians should acknowledge the importance of future fertility [16,20,24] and, by extension, the importance of FP counselling [8,24]. Studies indicated that FP counselling should be a routine procedure when AYAs are diagnosed with cancer and are at risk of infertility [8,29,33,56]. Patients also stated that they should be provided the opportunity to preserve fertility [53,58].

#### 3.4.2. Starting a Conversation about Potential Fertility Decline

With the acknowledgement of the importance of future fertility, the question is: when should the conversation about potential fertility decline be started? Studies show strong cross-gender support for starting the conversation about potential fertility decline due to gonadotoxic therapy early after diagnosis [20,24,28,37,53]. We need to realize that not all young cancer patients are aware of the potential fertility risks and may not even bring up the topic or ask for information [13,29]. Studies indicated that in most cases, a discussion about FP was initiated by the specialist treating the disease [13,44,45,55]. They reported that if a discussion about FP was initiated by the patient, they were already thinking parenthood or trying to conceive at the time of diagnosis [13]. Kayiira et al. mentioned that patients described that preparing to undergo cancer treatment was stressful and that there was no room to consider how the treatment would affect their future fertility. Therefore, the specialist always has to initiate the discussion [66]. In the study by Bentsen et al., however, most patients indicated that initial oncofertility counselling had not been offered by oncology specialists upon diagnosis or in subsequent consultations. The patients had to independently request specific information about FP. Some patients reported that they felt that in this situation, they were somewhat responsible for their own treatment regarding FP [8]. Bach et al. also stated that starting a conversation about this topic was difficult when a healthcare provider wanted the patient to focus more on their life-saving treatment [13]. With this lack of consideration, patients experienced dissatisfaction and regret. Insufficient initial oncofertility counselling had a huge impact on the experience of medical consultations during and after cancer treatment [8]. The absence of support and information from healthcare providers contributes to the psychological distress associated with potential fertility decline [39,71]. Crawshaw et al. noted that potential fertility decline should be discussed even if options for FP are neither available nor appropriate [37]. Anazodo et al. also reported that patients who had received FP information but did not start FP treatment valued how important this information was and reported feeling hopeful when given the opportunity to consider all aspects of life after cancer treatment at a very difficult time [33]. In a few studies, patients indicated that initiating the discussion was influenced by interpretation of social status (i.e., married, single, existing children) and that unwarranted assumptions are made about fertility desires and plans [24,27,33]. Also, (higher) age is seen as a factor for not offering FP services [9,39,52,54]. Studies, however, show that assumptions based on socio-demographic factors are not reliable determinants about a patient’s fertility desires and needs [24,32,64]. Therefore, Yee et al. recommended that physicians always address potential fertility decline and adopt a proactive approach to initiating the discussion [29]. Niemasik et al. concluded that routinely informing patients about potential fertility decline will help ensure that patients diagnosed with cancer are provided with the information they need to make an informed choice about their reproductive future [27].

#### 3.4.3. (Early) Referral for Fertility Specialist Counselling

After starting the discussion about potential fertility decline, studies emphasized the importance for patients to consult a reproductive specialist. Consultation by a reproductive specialist shows that fertility matters [29]. Anazodo et al. reported that consultations with a fertility specialist provided an opportunity to hear about different types of FP methods and to learn more about individual FP procedures and the success rates and complications associated with various FP procedures [33]. In the study by Ehrbar et al., patients who had not had a conversation with a reproductive specialist indicated that they would have liked to have a separate consultation with an expert to discuss FP [20]. Referrals to such specialists should be sent in a timely fashion. For men, as described by Yee et al., this is vital if they are to have the opportunity to bank sufficient sperm samples for future use [64]. In women, timely referral is even more important. Unlike infertile patients, women undergoing FP often lack any previous acquaintance with the subject of fertility and infertility. The urgent nature of the treatment means that information must be conveyed in a timely fashion to help women make the best possible decisions on how to proceed regarding FP [23]. To optimize oncofertility counselling, there is a need to bridge the gap between oncology and reproductive specialists [8]. Patients suggested an automatic referral to minimize deliberation in a situation in which a lot of information has to be processed [13,24].

### 3.5. The Need for Verbal and Written (Patient-Specific) Information

A clear need that emerged from the articles about FP counselling and treatment was the need for more information [13,16,20,28,31,36,44,45,53,55,65,66,69,70]. The need for information can be described in two ways: the need for the timely sharing of information and the need for (patient-specific) information.

The oncologist or surgeon is often the first person who informs the patient about a potential fertility decline. In the study by Vogt et al., patients specifically noted that they desired more information from their health care providers about risks, the effects of different chemotherapy regimens, and whether a delay would increase risks [69]. This was also seen in the study by Hill et al., who noted that little specific information about the potential effects of their systemic treatment on fertility was provided before the referral to a fertility clinic. Knowing beforehand would have allowed patients to focus on their options instead of on infertility statistics [44]. A need for the timely sharing of information was also reported by women receiving extensive information at their fertility consultation; they would have liked to have received this information earlier in their care pathway, as this would have enabled them to start thinking about FP options and questions to ask the fertility specialist [29,69]. Hill et al. also mentioned this. Those authors noted that several patients reported that they would have benefitted from written information about FP options and success rates before the FP consultation [44].

Many cancer patients have limited knowledge of FP techniques [16,53]. As such, there is a need to receive clear information [40]. Patients want to be informed about their fertility risks and FP options [21,29]. Armuand et al. also described that men had a positive experience, having received information from healthcare providers conveying a feeling of importance by encouraging them to bank [34].

Written information is valued as part of consultations with health care providers [21,69]. Bach et al. described that many patients explained that they were in a state of shock at the time of FP counselling and that this prevented them from fully comprehending the technicalities. [13]. Written information can be helpful in such situations. Ehrbar et al. indicated that with written information, a patient is likely to be better informed about FP; this also prevents patients from forgetting important information [20].

Information should be easy to access. Here, a problem was reported. Patients described difficulties in finding FP information specific to cancer patients [29]. Srikanthan et al. also described the need to improve resources and the delivery of information to patients [28]. Written information regarding cancer treatment and FP options either did not exist or was too generalized.

Other studies indicated that women wanted more information about their FP options, regardless of cancer diagnosis [69], and were pleased to have been offered options for FP, including doing nothing, and having the options and their implications explained to them [24]. Wang et al. described that some patients wanted access to additional information resources, especially information relevant to their specific situation or testimonies from other patients who had undergone FP treatment [41]. On the other hand, patients also reported feeling overwhelmed by the amount of information presented to them [41] and options that were irrelevant for them, making them insecure about whether the right choice had been made regarding FP [8]. Del Valle et al. [19] noted that information was a double-edged sword. On the one hand, the patients sometimes perceived a lack of information about the process of FP and they requested more explanations; on the other hand, too much information overwhelmed them and created more anxiety. Garvelink et al. also mentioned that women were ambivalent about the information they received about FP. They seemed positive, but they also mentioned negative characteristics, e.g., issues that remained unclear to them due to a lack information or gaps in information [21]. In the study by Ehrbar et al., patients reported feeling very clearly, sometimes too comprehensively, informed about the details of FP. A large number of patients stated that they would use an additional support tool in order to find validated, objective, structured, in-depth information as a means to receive counselling [70].

### 3.6. The Psychological Effects of Facing Potential Fertility Decline

Studies indicated that in AYAs, a cancer diagnosis is experienced as immense: to be diagnosed with cancer and challenged to think about your fertility after just being confronted with potential death [13,18,68]. In this process, AYAs position themselves as having limited agency due to factors that are outside of their control [39,68]. Zanagnolo et al. described that infertility has a strong potential to cause distress [57]. Bentsen et al. also reported that thoughts about reduced fertility became overwhelming and frightening [15]. Some studies indicated that the way in which information of potential fertility decline is experienced depends on the life stage at the moment of diagnosis of a patient. Not having children at time of diagnosis is associated with a greater likelihood of fertility concern [50,51]. Studies involving female patients reported that among females who wanted to have children in the near future, a potential fertility decline created considerable stress and anxiety [19]. The diagnosis derailed their plans, leading them to feel overwhelmed [28]. Women challenged to think about their fertility reported that this shook the foundation of their feminine identity [25]. In addition, besides just thinking about their fertility, many female patients had never considered the possibility that they would require fertility treatment, and as such, felt overwhelmed by the pressure to make a decision [41]. Wang et al. also reported that fertility discussions and access to FP may improve patients’ emotional health and minimize ongoing fertility concerns. This would allow patients to put concerns aside at the time of diagnosis [41].

Some patients felt it would be beneficial to receive additional psychological support and counselling. They suggested that this might help patients through the decision-making process and FP treatment during this overwhelming period [24,28,41,44]. Parton et al. also reported that females diagnosed with cancer at an older reproductive age may require particular support in terms of exploring fertility options and coming to terms with the outcomes for their fertility following cancer, combined with normal age-related fertility decline [39].

### 3.7. Undergoing Fertility Preservation

After initiating the discussion about potential fertility decline, the process of FP starts, i.e., counselling about FP options and the decision making about whether or not to start a FP treatment.

#### 3.7.1. Counselling

A few topics that emerged in research about experiences and needs regarding FP counselling and treatment were communication skills, organizational matters, and the cost of FP treatment.

##### Communication Skills

Multiple studies indicated that patients experience counselling as satisfying if a health care provider is proactive in informing the patient about the possible consequences of cancer treatment on fertility. The counselling should be informative, clear, and accurate [24,39,71]. Srikanthan et al. described that the process of decision making is experienced as a deeply personal choice and that all discussion should be sensitive to this [28]. Others added that the subject of FP counselling should be brought up with professional sensitivity, and the patient should have a choice in who is present during the discussion [8,33,37,58].

Kirkman et al. mentioned that crucial factors regarding communication were reported, i.e., being listened to and being treated with respect, no matter what the personal circumstances or desires are [24]. A health care provider should be open to hearing about personal aspirations [24]. In the study by Bentsen et al., patients also reported wanting to be met with understanding. They wanted to be taken seriously and reassured that a reproductive specialist would help as much as possible with FP and any subsequent fertility treatment. This would allow them to focus on the cancer treatment itself and convalescence [8]. Von Wolff et al. reported that counselling by a specialist about FP techniques is very satisfying to all women undergoing gonadotoxic treatment, irrespective of whether they decide for or against any specific FP treatment [56]. Canzona et al. reported that a critical turning point for FP decision making was the encountering of direct, supportive communication during initial fertility conversations with health care providers [36].

A specific point that emerged in a few studies about male AYAs was that health care providers should acknowledge that the procedure of sperm banking can be experienced as embarrassing [34,37,58]. Armuand et al. elaborated that they should give the patient the opportunity to consider who is present during the conversation about this topic. Also, the act of providing sperm in a clinical environment for fertility preservation can lead to distress, and offering alternatives, such as producing the sperm sample at home, may be helpful [34].

##### Organizational Matters

As mentioned before, studies reported a need for timely specialized fertility counselling and FP treatment. Inhorn et al. described that patients were grateful for well-coordinated and integrated oncofertility services with continuity of care between clinics [23]. Kirkman et al. also indicated that multidisciplinary care from oncologist, surgeons, and fertility specialists, but also nurse consultants, psychologists, and general practitioners, contributed to the quality of care [24]. It is considered helpful if the process is well organized and no organizational involvement is needed from the patient’s side [70]. Male AYAs experience FP more often as an integrated part of their cancer care [34]. Possibly because the procedure is relatively straightforward, men are more actively encouraged to consider sperm storage [9]. However, in the study van Bentsen et al., female AYAs often noted that they felt that they were falling between two different departments. They had the feeling that they had to facilitate communication about FP issues [8].

Wang et al. also reported that some patients felt out-of-place in the environment of a fertility treatment center due to their young age, single status, or male gender. They viewed these places as being primarily for females and as heterosexual couple fertility treatment centers [41].

Bentsen et al. also reported that patients expressed a wish to have an offer for consultation for their partner only at the fertility unit [8].

##### Costs of Fertility Preservation Treatment

Studies indicated that in countries where FP treatments are not covered by health insurers, the costs for FP can directly affect accessibility [28,35]. Costs were mentioned as a barrier for FP referral or FP procedures [30,52,67,69]. The costs that need to be considered are not just the financial burden of cancer (i.e., not being able to work) and the cost of medical treatment, but also the anticipated future costs of IVF using stored biological material [38,39,41]. In studies of Latif et al. and Yee et al. about sperm cryopreservation, costs seemed to be less of a barrier to undergoing FP treatment [32,64].

Patients for whom costs were prohibitive experienced higher decisional conflict [49]. Canzona et al. reported that difficulty weighing the decision to pursue FP because of costs is more prevalent in racial ethnic minority groups [36]. Anazodo et al. also mentioned that the financial aspect of FP or assisted reproductive treatment is a significant burden, leading to additional psychological distress and relationship problems [33].

Wang et al. and Walasik et al. stated that when FP treatments were not covered by insurers, patients felt that the costs of fertility treatments for cancer patients should be further subsidized [41,55].

#### 3.7.2. Decision Making about a Fertility Preservation Treatment

After receiving counselling about FP options, patients need to decide whether or not to start a FP treatment. Patients described the difficulty of having to make decisions rapidly. They have to make decisions under stressful circumstances, just after a cancer diagnosis [24]. Chapple et al. described that many young men felt rushed into making a decision about sperm banking at a time when they were overburdened with information and shocked by their recent diagnosis [31]. Being forced to make this decision added to the emotional burden of coming to terms with the prospect of cancer treatment. The emotional and physical burden of cancer sometimes resulted in having reduced capacity for decision making and, with that, reduced the likelihood of undergoing FP [39]. Garvelink et al., however, noted that for some women, FP was viewed as one of many decisions to be made, and while they were already in decision-making mode, it made it easier to decide about FP [21].

Baysal et al. revealed in their study that FP decision-making among young female patients scheduled for gonadotoxic treatment is mainly based on weighing two issues: the intensity of the wish to conceive a child (in the future) and the expected burden of undergoing FP treatment [43]. Women also reported a desire to avoid future regret [22]. Garvelink et al. noted the main reason for undergoing FP was to do everything to ensure future fertility [21]. Ethical and religious reservations are important around decision making. Ethical reservations are there especially with regard to the consequences of unused material and concerns that cancer therapy might not be effective [20]. Patients value being given the choice, opportunity, and time to fully investigate FP options [71]. Although the procedure of sperm cryopreservation is relatively straightforward, men also emphasize the importance of allowing time for decision-making [34].

The process of decision making about FP was positioned as a barrier, primarily by women, who would require a longer, more physically demanding FP procedure compared to men [39]. Some women also experience an ongoing tension between ensuring their own survival through cancer treatment and the desire for a biological child [39]. Especially women with hormone-sensitive breast cancer, confronted with the threat of increased hormone levels during ovarian stimulation, shared that they experienced a threat of possible cancer growth due to FP [18,69]. A less positive experience in the decision-making process was associated with higher decisional conflict, decisional regret and lower decisional satisfaction [42,48]. A higher quality decision is positively associated with a better experience in the decision-making process. The support of a health care provider is crucial for the decisional satisfaction of patients who decided not to pursue FP [48]. Marino et al. also reported that parental or partner involvement in decision-making was considered helpful [67].

#### 3.7.3. Experiences and Needs in Fertility Preservation Treatment

Patients that proceeded with FP treatment, reported that it is physically and emotionally challenging to have fertility treatment while simultaneously managing cancer.

In studies were experiences in FP treatment are described, a number of general topics emerged: sense of control, hope and future oriented, source of distress and the need for short term follow up.

##### Sense of Control

Patients describe a feeling of regaining a sense of bodily integrity and control through the reconstitution of reproductive choice [13]. It allowed patients to maintain a sense of control following the cancer diagnosis [22,39].

##### Hope and Future Oriented

Although men expressed fear for being infertile and experienced infertility as a loss of their manhood, they mentioned how good it was to know that they could have biological children in the future through the frozen sperm [30,34]. This sense of hope for conceiving a biological child in the future is an important aspect described by male and female patients after FP treatment was conducted [13,15,18,21,25,29,31,35,39,46,64]. It was also described as alleviating infertility related distress, thus allowing patients to feel more comfortable taking up cancer treatment [39,41,64]. Vogt at al. described that patients consider FP as positive, describing “peace of mind”, being able to “turn the negative of cancer into a positive” and “giving hope” [69]. Yee et al. similarly described that women who had cryopreserved embryos shared that this gave them hope for recovery and mental strength to fight cancer [29]. Cryopreserved material is seen as a type of insurance [13,16,23,41]. It presents an orientation towards the future. It gives patients a feeling of being directed towards survival. This is illustrated by a chance of parenthood and positive attitudes from healthcare providers generating a belief in survivorship and life after cancer [13,18,72]. Inhorn at al. documented that women who had at least one cycle of oocyte cryopreservation described this as a gift, blessing, miracle and form of empowerment [23]. Patients acknowledge however, that stored material is no guarantee for a future pregnancy [13,16,23] but maintaining the ability to conceive is significant for them [25].

##### Source of Distress

But FP can also be a source of distress. Dahhan et al. reported that the requirement of a cancer treatment shortly after FP causes an intense time pressure during FP [18]. This time pressure in FP was also mentioned by Bentsen et al. as “a race against time” while the cancer therapy had to start immediately [15]. Cordeiro Mitchell et al. noted that several patients described cryopreserved ovarian tissue as a sort of double-edged sword, providing hope as discussed above, but anxiety because of the uncertainty about the material (is a person going to be able to use it, will it work?) [16]. This source of distress was also mentioned by Salsman et al. in combination with an ongoing uncertainty regarding the fertility [40]. Del Valle et al. also reported that, due to the fact that the procedure of cryopreservation of ovarian tissue has to be carried out more quickly and more traumatically, the impact of the diagnosis was experienced as more intense [19].

In the research of Canzona et al. patients reported that FP is not always achieved and procedures they endure are uncomfortable and embarrassing [35]. Wang et al. also described the burden of FP treatment, disappointment and ongoing concerns, for example, a small number of oocytes able to be collected [41]. After failed FP attempts, Canzona et al. mentioned that patients are unsure future attempts will be successful [35]. Also distress about unknown fertility outcomes (no guarantee) and the burden of their partner (in particular female partners) potentially undergoing fertility treatment are described [39]. Single women confronted with a cancer diagnosis and FP are also worried about the fear of rejection by potential partners [17,25]. This is in line with the reported anxiety around current and future romantic relationships research of Canzona et al. They also reported sadness, guilt and jealousy surrounding friendships [35].

##### Short Term Follow Up Consultation after Fertility Preservation

Bach et al. described that in a crisis and information overload at the point of diagnosis, patients reported limited recollection or understanding of information received at the initial counselling [13]. Also in the study of Ehrbar et al. patients stated to be felt overwhelmed by the immense amount of information. The majority mentioned that it would be helpful to know that reproductive health can be revisited later [70]. Patients in the study of Yee et al. also indicates that follow up after FP was important. It provided in-depth information about sperm quality and better understanding of the results prior to the start of oncological treatment [64].

### 3.8. Worries around Future Fertility and Increase Cancer Risks

Patients’ desire for future father- or motherhood are main determinants associated with undergoing FP treatments [64]. Anazodo et al. mentioned that many patients reported more anxiety about their fertility potential after cancer treatment than at the time of diagnosis and FP [33]. Walasik et al. described that patients were mostly concerned about the safety of having children after oncological treatment [55]. Canzona et al. stated that patients have worries and uncertainty that cancer and/or treatment makes it less likely for them to be healthy enough to raise future children [35]. Despite anxieties about surviving to see their children growing up, 24% of childless men felt that having cancer had increased their wish to have children [62]. Wang et al. and Canzona et el. reported concerns about the impact of chemotherapy on the consequent health of future children [35,41]. Schover et al. stated that 31% of the patients in their research believed that their children would definitely be at increased risk for cancer. A smaller percentage, about one fourth worried that their past cancer treatment could affect the health of children conceived afterward [62]. Zhang et al. [65] and Achille et al. [30] also stated the concern about the possibility of transmitting a disease to their progeny. Women carrying BRCA mutations were more likely to have increased concern about future children inheriting increased cancer risk. This highlights the importance of incorporating tailored, risk-mitigating recommendations into fertility counseling [47,51,69]. In women with estrogen-sensitive breast cancer there was the concern of how to achieve a safe pregnancy in the future [41,69]. This safety concern was also mentioned in association with risks involved with ovarian tissue transplantation. Ovarian tissue (re)connects the patient with their disease in ways that frozen oocytes or embryos do not. Worries about risks in post-cancer reproduction were especially prevalent among women who had estrogen-sensitive breast cancer, those who had tested positive for BRCA genes and those, who had for instance, sarcomas in the lower parts of the body [13]. Another concern mentioned is about not being able to take care of their children in case of disease relapse [51].

A different concern emerged is the effect of cancer diagnosis on psychosocial aspects. Anazoda et al. mentioned that the fear of being infertile had a negative impact on starting intimate relationships [33]. In the research of Salsman et al. patients reported that a cancer diagnosis and potential infertility would make them less desirable partners. Preserving their fertility was a priority to mitigate those fears [40].

### 3.9. Follow Up after Fertility Preservation

When FP treatment has been completed, consultation with the fertility specialist is, most of the time, ended. However, patients want health care providers to be aware of and discuss the impact of chemotherapy and infertility after completion of active treatment. The negative impact of chemotherapy on quality of life during survivorship remains important. Respondents endorsed limited discussions with health care providers about how and when to engage with fertility specialists after chemotherapy, and possible fertility screening or surveillance that can be undertaken [14,28]. Bentsen et al. compared this subsequent waiting time to a marathon. Not until the waiting time was over, the participants found out the consequences of the cancer treatment [15]. Many studies list the importance of informational follow ups to lower distress in patients. Patients experience distress when they have less knowledge about reproductive biology. Patients want consultation regarding their fertility concerns and report uncertainty about the time range for fertility treatment after cancer therapy. Patients also notify that they were in doubt whether they still belonged to the fertility unit or not, and where to obtain information. Distress can also be caused by questioning whether to continue the storage of the preserved material [8,13,16]. Bentsen et al. reported that it is advised to offer the possibility of fertility assessment after cancer treatment [8]. Benedict et al. stated that provider-initiated discussions relieved patients form the burden of bringing up concerns themselves. Patients worry about missing critical information or reproductive time window and fear early menopause [14]. Bach et al. also reported a need for interventions to handle and reduce fears regarding risks of re-transplanting ovarian tissue and post-cancer reproduction within a clinical care pathway of fertility preservation and post-cancer reproduction [13]. Patients in the research of Ehrbar et al. clearly stated interest in aspects beyond FP, such as fertility and contraception, sexuality, masculinity, and impact on couple life [70]. Anazodo et al. reported that FP consultation in the survivorship period is seen as an opportunity to talk about sexual health, safe sex practice and symptoms of sexual dysfunction and to receive advise and support about fertility-related psychosocial distress [33]. Benedict et al. also reported a need for emotional support in post-treatment care where counselling can help with uncertainty and distress [14]. Kirkman et al. reported to be alert to the need for continuing psychological care as women confront the fear of recurrence and grief about lost fertility [24].

## 4. Discussion

This systematic review was conducted to get an overview of patient reported outcomes (PROs) and patient experiences regarding the counselling, treatment and future fertility in FP in adolescents and young adults (AYAs) with cancer. Various studies report on PROs and patient experiences in FP in AYAs in the intense situation of being diagnosed with cancer but also being confronted with potential risk of infertility due to the necessary treatment. Relevant PROs and patient experiences with FP counselling are summarized in Table 2 and include a proactive approach in initiating the conversation about potential fertility decline by oncological specialists when there is a need to start a gonadotoxic cancer treatment. All AYAs should have the opportunity for an open discussion about the possibilities for FP and early referral to a fertility specialist is essential. In addition, patients report the need for additional patient specific information, emphasizing the value of verbal and written patient information about FP. Finally, there is a need for follow up in these patients after the FP treatment has been completed.

The acknowledgement of the importance of future fertility was also confirmed in the review of Taylor et al. where they defined this PRO as “fertility in trust” as an obligation of the health care provider to recognize the long-term importance of fertility. Here the long-term effect on fertility is influenced by short term decisions [73]. In addition, some AYA believe the possibility of FP is an expression of professional belief that they have a future [37].

This review shows a need for clear information provision in the whole process of counselling, treatment, but specifically in follow up after FP counselling (and treatment). The need for information is mentioned in reviews of Daly et al. [74] and Linnane et al. [75] on factors affecting patient FP decision making, where information provision was often perceived as inadequate or unclear. Recently, Clasen et al. also mentioned the under reporting of regrets and concerns after FP counselling, possibly explaining the variable satisfaction with fertility information [76]. The focus of these reviews, however, was narrowed to the decision-making process in FP. Our review informs that the need for information is more extensive. Where consultation with a fertility specialist after FP counselling and FP treatment most often ends, patients show specific needs for further follow up. The information overload after a recent cancer diagnosis demands later revisiting and after a cancer treatment more information concerning future fertility is needed.

The need for FP follow up care is also mentioned by Gonçalves et al. They focused on perspectives of FP in young women with gynecological cancer and described the need for follow up care [77]. Macklon and Fauser concluded that this follow up after FP is important; issues related to the use of the stored material could be addressed at these visits as well as safety concerns that some of the patients may have [78].

Of course, results of included articles in our review could be influenced by different factors. First, experiences in FP are possibly influenced by the country of residence. Accessibility of oncofertility services can be influenced by multiple factors, including financial aspects of FP treatment [70]. In a country where there is no reimbursement, costs of FP treatment could be an obstacle. Our review showed that not all studies indicated that costs are a factor in FP decision making and FP treatment. This was however reported in studies regarding sperm cryopreservation [32,64]. In this treatment costs to conduct FP are relatively low. Selection bias is questioned where included patients already underwent sperm cryopreservation [64]. Costs for cryopreservation of oocytes on the other hand are considerably higher and absence of reimbursement could have a great effect on the decision regarding FP. In our review a large amount of included studies are conducted in high income countries. This could possibly lead to a narrow view and with this missing of other patients’ specific needs in FP counselling and FP treatment.

Secondly, the year of performed FP counselling and treatment of included patients in a study is also of importance. The field of FP options has grown hugely in the last years. On one hand because of the increasing recognition of the importance of potential loss because of gonadotoxic treatment, and on the other hand because of increasing technologies of preservation options. Where ovarian tissue cryopreservation and transplantation was described as a scientific treatment ten years ago, nowadays it is a standard form of care. And where embryo cryopreservation was, a long time, the standard procedure for FP in women, with no opportunity for patients without a male partner, this has been changed with the introduction of vitrification of oocytes over the last decade. In addition, the proportion of FP discussion and fertility specialist consultation has changed with the introduction of newer methods [52]. Finally, with the embedding of FP counselling and treatment in oncological protocols, referral patterns have changed over time towards better accessibility of FP for patients. In this context we also have to take into account that differences in experiences of patients could have been influenced through timing of the treatment. Peddie et al. for example described that women were feeling negative about FP because they didn’t have the opportunity due to less FP possibilities at that time [9]. Unfortunately 40% of the included studies in this review didn’t describe the year of FP counselling and treatment and since others have a wide distribution in year of FP counselling and treatment it is difficult to show differences in PROs and patient’s experiences in FP over time between the different studies.

Lastly, there were differences in patient populations between the studies. Some studies included only patients with the same cancer diagnosis, others had multiple cancer diagnoses in their inclusion. Another difference between the studies was the time between diagnosis and performed examination. This time varied between a couple of weeks up to more than 25 years. Next to gender and with that potential FP options, all these differences could possibly influence the reported outcomes and experiences.

In this study we have decided to focus on AYAs experiences. Of course, FP treatments are also available and needed for prepuberal children or (male) patients ≥40 years. However, we decided to exclude these patients since other aspects, such as diminished ovarian or sperm quality, could interfere with the possibilities of FP. Specifically, in prepuberal girls and boys FP possibilities are complex or experimental. Moreover, patient reported outcome measures of FP can be difficult to define with these young patients and also possibly the influence of their parents. Still, talking about potential fertility decline and options of FP is essential in these groups [79].

## 5. Conclusions

Being confronted with a potential life-threatening disease and simultaneously have to consider FP treatment is an intense situation for AYAs diagnosed with cancer. In this review we summarized the published PROs and patient experiences regarding the counselling, FP treatment and future fertility. This includes the need for patients to acknowledge the importance of future fertility, more patient specific information and the need for follow up after oncological treatment that has a risk of fertility decline.

As mentioned by del Valle et al. despite advances and increasing awareness about the importance of the integral treatment of cancer and FP, there is a lack of knowledge regarding patient experiences and needs in this process [19]. We believe a clear FP pathway can prevent delays in receiving a referral to a fertility specialist to discuss FP options and initiating FP treatment. By measuring the patient reported outcomes and patient reported experiences (PROMs and PREMs) and incorporating these in a FP pathway, experiences around FP will be optimized and a process established to achieve long-term follow up after FP treatment.

## Figures and Tables

**Figure 1 cancers-15-05828-f001:**
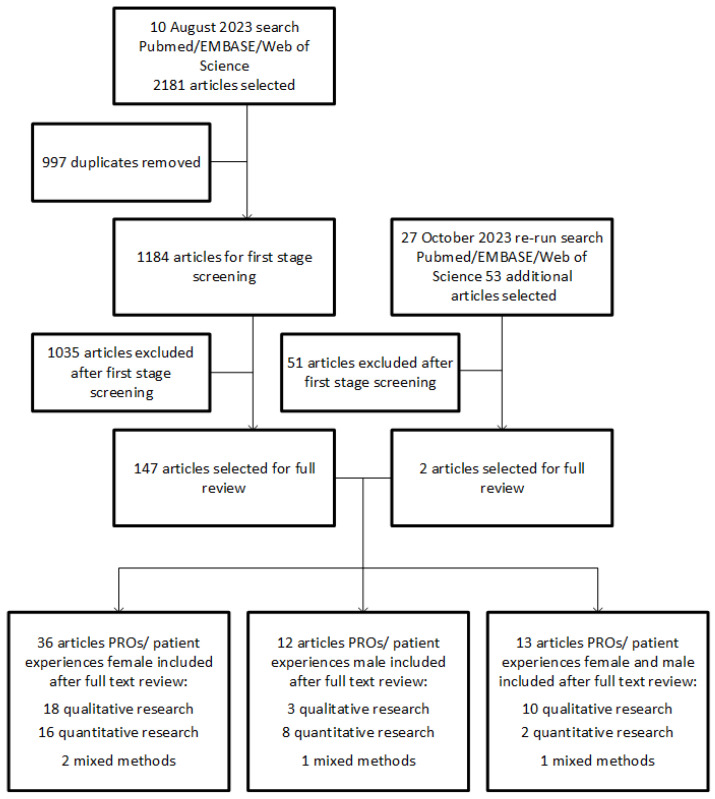
PRISMA flow diagram of the study selection process.

**Table 1 cancers-15-05828-t001:** Characteristics of included studies.

Study	Origin	Year	Aim	Type	Study Design	Study Period	Eligibility/Inclusion Criteria	Year of FP Counselling	Sample	Data Collection	Relevant Finding
Female											
Bach et al. [13]	Denmark	2020	Explore patients’ experiences with ovarian tissue cryopreservation (OTC), including their reflections on the long-term storage of tissue and the use of surplus tissue.	Qualitative	Retrospective	2017–2019	Underwent OTC in Denmark between 2003 and 2018.	2003–2018	*N* = 42 (age range at time of interview: 22 years—early 50 s), 32 of whom had ovarian tissue transplanted (age of OTC: 15–42 years); Unclear follow up time. Multiple diagnoses, i.a. breast cancer and lymphoma	Semi-structured interviews	OTC can be regarded as a ‘hope technology’, but in contrast to the freezing of oocytes and embryos, ovarian tissue is interlinked with scenarios of risk and disease connected to tissue transplantation. It is perceived as a future means to provide options beyond the scope of reproduction. There is a need for the specialized and sensitive provision of information, regular follow-up, and fertility counselling after OTC and cancer treatment
Bastings et al. [42]	The Netherlands	2014	How do female patients experience FP consultations with a specialist in reproductive medicine, and how does this influence subsequent FP decision-making?	Quantitative	Retrospective	2013	Aged ≥ 16 years when the study was conducted. Had undergone FP after FP counselling.	2008–2013	Response rate: *N* = 60/108 patients (56%) who returned completed questionnaires; Mean age: 29 years; Mean follow up: 2 years; Multiple diagnoses, i.a. breast cancer (60%) and lymphoma (18%)	Study specific questionnaire and Decisional Conflict Scale (DCS) and Decisional Regret Scale	The majority of patients were satisfied with FP counselling. Some patients wished for more information about specific subjects (e.g., the influence of hormones and ovarian enlargement on the risk that chemotherapy would cause ovarian failure). Negative experiences were found to be associated with decisional conflict and decision regret.
Baysal et al. [43]	The Netherlands	2015	To identify the issues that women consider to be important in their decision-making process, at the time of FP counselling, about whether or not to undergo FP in a setting where financial factors do not play a role. Furthermore, to investigate how these issues are related to patients’ FP choices.	Quantitative	Retrospective	2013	Age ≥ 16 years, not severely diseased, no psychological problems. Received FP counselling and if at least one FP option was offered after FP counselling.	1999–2013	Response rate: *N* = 87/143 patients (61%) who returned completed questionnaires (49 who chose to undergo FP, 38 of whom refrained from FP options). Mean age at counselling: 28 years; Mean follow-up time for the total group of responders: 3 years (SD 2); Multiple diagnoses, i.a. breast cancer (58%) and lymphoma (18%)	Study specific questionnaire	FP decision-making in young women scheduled for gonadotoxic therapy is mainly based on weighing two issues: the intensity of the wish to conceive a child in the future and the expected burden of undergoing fertility preservation treatment.
Benedict et al. [14]	USA	2021	To explore survivors’ recommendations in terms of addressing fertility and family-building needs after cancer.	Qualitative	Retrospective	Unclear	Female cancer survivors aged 15–39 years, completion of gonadotoxic treatment, had not had a child since cancer diagnosis and reported parenthood desires or undecided family-building plans.	Unclear	*N* = 25; Mean age: 29 years; mean age at diagnosis: 23 years. Multiple diagnoses, i.a. Hodgkin lymphoma (24%), breast cancer (20%), and leukemia (12%)	Semi-structured interviews	Six primary recommendations were identified for post-treatment care. Health care providers should offer more information about fertility and family building. Providers should not make assumptions about patient’s family-building desires and intentions. Emotional support should be offered. There is an overarching need for guidance about how to translate information into actionable next steps. Improve communication. Provide financial information and refer to peer support resources.
Bentsen et al. [8]	Denmark	2021	To examine how female AYA cancer patients and survivors experienced initial and specialized oncofertility counselling and to present their specific suggestions on how to improve oncofertility counselling.	Qualitative	Retrospective	2020	Female cancer patients and survivors aged 18–39 years.	Unclear	*N* = 12 patients, 20–35 years; Mean age: 28 years; Patients in treatment (*N* = 4) and post treatment (*N* = 8); Unclear follow up time. Multiple diagnoses, i.a. ovarian cancer (25%) and non-Hodgkin’s lymphoma (25%)	Semi-structured interviews	There is a continuing problem regarding insufficient oncofertility counselling to AYAs with cancer. There is a need for further improvement to ensure uniform and adequate information, especially in initial oncofertility counselling. Patients suggest focus on verbal and written information, along with communication upskilling to improve oncofertility counselling
Bentsen et al. [15]	Denmark	2023	To explore thoughts about fertility among female AYAs with cancer.	Qualitative	Retrospective	2020–2021	Female AYAs and cancer survivors (18–39 years).	Unclear	*N* = 12; Mean age; 28 years; Multiple diagnoses, i.a. Hodgkin’s lymphoma (25%) and breast cancer (17%)	Semi-structured interviews	Four main themes were found: Female AYAs held on to the hope of having children in the future, female AYAs experienced time pressure and waiting times as a sprint as well as a marathon, female AYAs faced existential and ethical choices about survival and family formation, and female AYAs felt a loss of control of their bodies
Cordeiro Mitchell et al. [16]	USA	2020	To determine the extent to which patients understand the potential fertility and medical benefits of the cryopreservation of ovarian tissue, the desire for future fertility, and feelings that patients have about the cryopreservation procedure.	Qualitative	Retrospective	Unclear	Age ≥ 18 years, had undergone ovarian tissue cryopreservation.	2006–2017	*N* = 8/9; Age at interview: 19–37 years; Mean follow up time: 5 years. Multiple diagnoses: hematologic or gynecological cancer	Telephone semi-structured interviews	There is a knowledge gap among patients with cryopreserved ovarian tissue regarding the uses and benefits of OTC. There is also a strong desire among these patients for improved education about this technology.
Corney et al. [17]	UK	2014	To focus on the stressors and vulnerabilities faced by young, single, childless women who were diagnosed with a first episode of breast cancer.	Qualitative	Retrospective	Unclear	Single, childless women with a first episode of breast cancer.	Unclear	*N* = 10; Age ranges: 27–41 years at diagnoses, 30–44 years at interview; Time range since diagnosis: 8 months–5 years; Single diagnosis of breast cancer	Semi-structured interviews	Young, childless single women with breast cancer face additional vulnerabilities and may benefit from tailored support from health care providers and interventions specifically targeted at them.
Dahhan et al. [18]	The Netherlands	2021	To explore how women experience oocyte or embryo banking when they have just been diagnosed with breast cancer.	Qualitative	Retrospective	2013–2014	Women aged 18–43 years, newly diagnosed with breast cancer and who banked their oocytes of embryos in two Dutch university medical centers.	2013–2014	*N* = 21/28; Mean age: 32 years; On average, 8 months after fertility preservation; Single diagnosis of breast cancer	Semi-structured interviews	Three main experiences: the burden of FP, the new identity of a fertility patient, and coping with breast cancer through FP.
Del Valle et al. [19]	Spain	2022	To identify cancer patients’ specific needs and experiences regarding FP.	Qualitative	Retrospective	2019	Females of reproductive age (18–44 years) diagnosed with cancer that could affect reproductive function due to the need to receive gonadotoxic treatments, who had undergone treatment for FP.	2017–2019	Response rate: *N* = 14/24 (58%); Mean age = 32 years (SD: 5); Time since FP was between 8 and 20 months; Multiple diagnoses, i.a. breast cancer (57%) and lymphoma (14%)	Semi-structured interviews	In the ovarian tissue freezing group, feelings, emotions, stress, and the impact of the diagnosis were more intense due the fact that this procedure has to be carried out quickly, making it more traumatic. Patients suffered from difficulties when making decision about fertility whilst dealing with a cancer diagnosis. They needed adequate information and support from health care providers. Despite increasing awareness of FP, there is a lack of knowledge regarding patient experiences and needs related to this process.
Ehrbar et al. [20]	Switzerland	2016	To assess the significance of fertility issues in cancer patients, their attitude toward fertility preservation, potential decisional conflicts, and patients’ needs during the decision-making process.	Qualitative	Retrospective	2012	Female cancer survivors aged between 18 and 45 years, had a cancer diagnosis within the last 10 years with a treatment that might have affected their fertility.	1999–2011	Response rate: *N* = 12/21 (57%); Age range: 21–45 years at time of study; Mean time since diagnosis: 5 years; Multiple diagnoses, i.a. breast cancer (67%)	4 focus groups	The significance of fertility was high and attitudes toward FP were positive. Religious and ethical reservations were not negligible. More support was desired and specific tools would be beneficial.
Garvelink et al. [21]	The Netherlands	2013	To describe the experiences of women who had received at least one counselling consultation on FP in relation to information provision and decision making about FP.	Qualitative	Retrospective	2007–2008	Women who had at least one counselling consultation about FP, between 18 and 40 years of age at the time of counselling.	2002–2007	Response rate: *N* = 34/53 (64%); Mean age: 33 years at time of interview, 31 years at FP consultation; Mean time since counselling: 24 months; Multiple diagnoses, i.a. breast cancer (82%), non-Hodgkin lymphoma (6%), and Hodgkin lymphoma (6%)	Semi-structured interviews	Women recommended standardization of information provision, improvement of communication, and availability of FP-specific patient information materials to improve future information provision processes. Overall, women were satisfied with the timing and content of information provision, but women were less positive about the need to be assertive to obtain information and the multiplicity of decisions and actions to be carried out in a very short time frame.
Hershberger et al. [22]	USA	2016	To help understand young women’s reasons for accepting or declining FP following a cancer diagnoses.	Qualitative	Retrospective	Unclear	Female cancer patients aged between 18–42 years, eligible for FP. Made a decision regarding FP within the past 18 months.	Unclear	*N* = 27, of which 14 declined FP, 13 accepted FP; Mean age: 29 years; Average: 5 months prior to study diagnosed with cancer; Multiple diagnoses, i.a. breast cancer (52%), Hodgkin’s lymphoma (19%), and ovarian cancer (15%)	Study specific questionnaire interviews by telephone (*N* = 21) or email (*N* = 6)	The primary factor upon which many patients-based decisions related to FP was whether the immediate emphasis of care should be placed on surviving cancer of securing options for future biological motherhood.
Hill et al. [44]	Canada	2012	To gather information from young patients with breast cancer about their experiences with FP referrals, consultations, and decision making.	Quantitative	Retrospective	Unclear	Patients with breast cancer who attended an FP consultation.	2005–2011	Response rate: *N* = 27/53 (51%); Mean age at diagnosis: 31 years; No clear follow up time; Single diagnosis of breast cancer	Multiple choice and open-ended survey	FP referral should be initiated by the surgeon as soon as a diagnosis of invasive cancer is made; women need written materials before and after FP consultation; and a FP counsellor who is able to spend additional time after the consultation could help with decision making.
Inhorn et al. [23]	USA/Israel	2018	To examine women’s motivations and experiences, including their perceived need for patient-centered care following the diagnosis of a life-threatening illness.	Qualitative	Retrospective	2014–2016	Women who had completed at least one medical egg freezing cycle at one of the six participating IVF clinics (4 USA/2 Israel).	2000–2016	*N* = 45 (33 USA/12 Israel), including 35 patients with oncologic indication; Follow up time not clear; Multiple diagnoses, i.a. breast cancer (43%)	Semi-structured Interviews	Special needs of medical egg freezing patients who tend to be young (<30 years), unmarried, and resource-constrained, making them highly vulnerable. Facing the frightening double jeopardy of cancer and fertility loss.
Kirkman et al. [24]	Australia	2013	To learn from women about their experiences of cancer care in relation to their fertility, to consider their recommendations to clinicians, and ultimately, to inform and enhance the provision of supportive care to such women.	Qualitative	Retrospective	Unclear	Women diagnosed with breast cancer, 18–45 years of age, and at least 1-year postdiagnosis	Unclear	*N* = 10; Age range: 26–45 years at interview (age range at diagnosis was 25–41 years); Follow up time not completely clear (based on age difference: 1–7 years); Single diagnosis of breast cancer	Semi-structured interviews by phone (*N* = 9) or in person (*N* = 1)	Fertility was important to participants. Complex psychological needs arising from the ramifications of living with cancer. Clinicians should be aware of the significance of fertility and avoid making assumptions based on the women’s age, marital status, or other characteristics. Women valued referral to a fertility specialist at the earliest opportunity and valued multidisciplinary treatment.
Ko et al. [45]	China	2023	To assess the knowledge, perceptions, and intentions regarding FP among women diagnosed with breast cancer.	Quantitative	Retrospective	2020–2022	Women diagnosed with breast cancer, 18–45 years or age.	Unclear	Response rate *N* = 410/461 (89%); Mean age at questionnaire: 40 years; Follow up time: less than five years for 72% of subjects. Single diagnosis of breast cancer	Study specific questionnaire	Younger age and higher educational level were significantly associated with increased awareness of FP. Awareness and acceptance of different FP methods was generally low.
Komatsu et al. [25]	Japan	2014	To explore the experience of undergoing a radical trachelectomy from the perspective of women with cervical cancer.	Qualitative	Retrospective	2011–2012	Women diagnosed with cervical cancer who had undergone radical trachelectomy (between 2006–2010) and who were encouraged to attempt conception after 6 months without cancer recurrence.	2006–2010	*N* = 15; Mean age at time of surgery: 32 years; No clear follow up time; Single diagnosis of cervical cancer	Semi-structured interviews	Women who undergo radical trachelectomy experience an identity transformation process. FP repairs the threatened feminine identity and keeps a window of hope open.
Komatsu et al. [26]	Japan	2018	To understand how women with breast cancer receiving FP counselling make fertility-related decisions.	Qualitative	Retrospective	2016	Women who were diagnosed with breast cancer and received FP counselling. Patients with strong physical discomfort or depression were excluded	2010–2014	*N* = 11; Mean age: 41 years at interview (SD 4 years). No clear follow up time; Single diagnosis of breast cancer	Semi-structured interviews	After receiving FP counselling, women with breast cancer made difficult decisions in stressful situations without sufficient healthcare information and support. Healthcare providers should be aware of and understand the unmet needs of women. Tailored information should be given to individual women in collaboration between oncology and reproductive health providers to support them in maintaining hope and a positive mindset throughout the decision-making process about fertility issues.
Leflon et al. [46]	France	2022	To evaluate experiences and gynecological and reproductive health outcomes in women aged over 18 years at the time of FP who had undergone OTC prior to receiving moderate or highly gonadotoxic chemotherapy or radiotherapy for malignant or non-malignant disease.	Quantitative	Retrospective	2019–2021	Women who had undergone OTC (>18 months ago) before receiving moderate or highly gonadotoxic chemotherapy or radiotherapy. Aged over 18 at OTC.	2004–2018	Response rate: *N* = 64/87 (74%); Mean age at OTC: 30 years; Mean age at questionnaire: 35 years	Study specific questionnaire. All the questions had multiple-choice answers with a free-text option for additional explanatory notes	Young adult women expressed a good satisfaction rate with OTC. The majority of patients thought that OTC had a positive impact on their well-being during disease treatment. A reassuring effect on their future fertility was noted, as was the hope of having a child despite gonadotoxic treatment
Lewinsohn et al. [47]	USA and Canada	2023	To describe future desire for biological children, concerns about fertility, and the use of FP strategies among carriers and noncarriers.	Quantitative	Retrospective	2006–2016	Women newly diagnosed with breast cancer, aged ≤ 40 years; stage 0–IV breast cancer <6 months before enrolment.	2006–2016	Response rate: *N* = 1052/1302 (81%); 118 positives for germline pathogenic variant in cohort; Median age at diagnosis: 36 years in the carrier cohort, 37 years in the noncarrier cohort; Single diagnosis of breast cancer	Study specific questionnaire	A breast cancer diagnosis may alter some young women’s desire for children and give rise to concerns about future infertility, but this does not appear to be impacted by mutation status. The high level of reported concern about future children inheriting cancer risk among carriers highlights the importance of incorporating tailored, risk-mitigating recommendations during fertility counselling.
Melo et al. [48]	Portugal	2018	Understanding patients’ perceptions about the FP decision-making process and quality. To examine the association between the patients’ perceptions of healthcare providers’ support in the FP decision-making process and FP decision quality, and to determine whether this association differed based on the FP decision.	Quantitative	Prospective	2013–2016	Female patients aged between 18 and 40 years; recent diagnosis of cancer and a need to undergo gonadotoxic cancer therapy.	2013–2016	Response rate: *N* = 82/110 (76%) T1 (directly after FP counselling); Mean age: 31 years; Response rate: *N* = 71/82 (87%) T2 (after cancer treatment). Multiple diagnoses, i.a. breast cancer (75%)	Study specific questionnaire. At T1 patients’ perceptions of the FP decision-making process were assessed. At T2 the patients’ current perceptions of the decision-making process, their decisional regret and satisfaction with the decision were also included.	A less positive experience in the decision-making process was associated with higher decisional regret and lower decisional satisfaction. A higher quality decision was positively associated with a better experience in the decision-making process. The support of a health care provider is crucial for the decisional satisfaction of patients who opt not to pursue FP.
Mersereau et al. [49]	USA	2013	To identify factors associated with decisional conflict regarding FP using data from a prospective cohort study of reproductive outcomes in female young adult cancer survivors.	Quantitative	Cross sectional	2011–2012	Women aged 18 to 44 years at the time of study enrolment with a personal history of cancer.	Unclear	*N* = 208; Median age: 31 years at study survey; Median time since diagnosis: 2 years; Multiple diagnoses, i.a. breast cancer (32%) and Hodgkin’s lymphoma (20%)	Study specific questionnaire and a modified version of the DCS	Increasing access to FP via referral or counselling and cost reduction may decrease decisional conflict about FP for young patients struggling with cancer and fertility decisions.
Niemasik et al. [27]	USA	2012	To determine what women recalled about reproductive health risks (RHR) from cancer therapy at the time of cancer diagnosis in order to identify barriers to reproductive health counselling and FP.	Qualitative	Retrospective	2010	Women aged 18–40 years with a diagnosis of leukemia, Hodgkin’s disease, non-Hodgkin’s lymphoma, or breast or gastrointestinal cancer.	1993–2007	Response rate: *N* = 1041/2532 (41%) completed the survey and 697/2532 (28%) responded to the open-ended question; Mean age at diagnosis: 32 years; Mean age at survey: 41 years; Mean time since diagnosis: 10 years	Study specific questionnaire	Many women may not receive adequate information about reproductive health risks or FP at the time of their cancer diagnosis. Advancements in reproductive technology and emerging organizations that cover the financial costs of FP have dramatically broadened the options women have to preserve their fertility. Routine and thoughtful counselling, as well as collaborative cancer care, will help ensure that women diagnosed with cancer are provided with the services and information they need to make an informed choice about their reproductive future.
Ruddy et al. [50]	USA	2014	To better understand the burden of concern about fertility, how fertility concerns affect treatment decisions, and the FP strategies used by women in a large cohort of young women newly diagnosed with breast cancer.	Quantitative	Cross-sectional	2006–2012	Aged ≤ 40 years and diagnosis with stage 0 to IV breast cancer <6 months before enrolment between 2006 and 2012.	2006–2012	*N* = 620/1511 (41%); Median age: 37 years; Median time between diagnosis and return of baseline survey: 141.5 days.	Study specific questionnaire, and the modified Fertility Issues and Outcomes Scale (FIS) and the Hospital Anxiety and Depression Scale (HADS)	Many young women with newly diagnosed breast cancer have concerns about their fertility, and for some, these substantially affect their treatment decisions. Not having children at diagnosis is associated with a greater likelihood of fertility concern
Ruggeri et al. [51]	Italy and Switzerland	2019	To characterize the experience of breast cancer in a cohort of young European women, focusing especially on fertility concerns and their impact on treatment decision making.	Quantitative	Cross-sectional	2009–2016	Aged ≤ 40 years with stage I-IV breast cancer, diagnosed < 6 months before enrolment (2009–2016).	2009–2016	*N* = 297/349 (85%) (207 from Italy, 90 from Switzerland); 32% of the women were aged < 35 years.	Study specific questionnaire a modified Fertility Issues Survey	Young women with newly diagnosed breast cancer have fertility concerns, including fear of increasing their personal or offspring cancer risk and not being able to take care of their children in case of disease relapse. In a multivariable analysis, having children was the only variable associated with fertility concerns.
Sauerbrun et al. [52]	USA	2023	To identify the prevalence and nature FP discussions and barriers to FP care.	Quantitative	Retrospective	2019–2021	Aged 18–42 years at time of breast cancer diagnosis.	2006–2016	*N* = 69/80 (86%) successfully contacted (322 eligible patients); Mean age at diagnosis unclear (70% between 35 and 42 years); Single diagnosis of breast cancer	Study specific questionnaire	Older women and those who were parents at the time of diagnosis were less likely to engage in a FP discussion. The most common reasons for declining FP consultation were already having their desired number of children, financial barriers, and concern about delaying cancer treatment and cancer recurrence. There was a higher proportion of FP discussions and consultation with a reproductive endocrinology and infertility specialist after 2013, when oocyte cryopreservation became non-experimental.
Schlossman et al. [68]	USA	2023	To identify the major barriers premenopausal individuals face in terms of accessing fertility care at the time of diagnosis with a gynecologic cancer, to assess patient experiences, particularly concerning fertility, and to learn how to better support such patients in the future.	Mixed methods	Retrospective	Unclear	Patients seen for follow up for ovarian, endometrial, or cervical cancer with no active cancer treatment. Aged 18–40 years at the time of diagnosis.	2012–2022	*N* = 55/228 (24%) completed questionnaires; Median age at diagnosis: 32 years; Multiple diagnoses of ovarian cancer (36%), endometrial cancer (22%), or cervical cancer (42%); *N* = 20 interviews; Median age at diagnosis: 32 years; Multiple diagnoses of ovarian cancer (50%), endometrial cancer (30%), and cervical cancer (20%)	Study specific questionnaire and semi-structured interview	Patients reported the emotional response of their diagnosis as a barrier to receiving fertility care, reporting lack of control and feelings of shock and confusion. Patients also identified inadequate counselling, a lack of time, economic constraints, and prioritization of cancer treatment as barriers.
Srikanthan et al. [28]	Canada	2019	To improve the information tools and decision aids available to women and to better understand patient experiences to improve the delivery of care. To understand patients’ experiences and attitudes regarding the delivery of care, fertility discussions, and FP at the time of initial diagnosis.	Qualitative	Retrospective	2014	Female breast cancer survivors, 39 years of age or younger at the time diagnosis, within 2 years of diagnosis (2012–2014).	2012–2014	*N* = 50/58 (86%); Median age: 35 years (range 25–39 years)	Semi-structured interview and four additional questions to elicit and quantify participant opinions on FP.	Six common themes: Requirement of more patient support; Improving information; Integration of patient values; Creating options for patients; Financial limitations; and The need to look beyond the immediate impact.
Urech et al. [53]	UK, USA, Switzerland, Germany and Austria	2018	To assess levels of knowledge concerning FP techniques and levels of confidence in that knowledge and attitudes toward FP. To assess differences concerning knowledge and attitudes between different language groups and healthcare systems and differences between participants who made use of any FP option compared to those who did not.	Quantitative	Retrospective	Unclear	Female cancer survivors aged ≥ 18 years who had a cancer diagnosis within the last 10 years and had a cancer therapy potentially affecting their reproductive function.	Unclear	*N* = 155 (80 English speaking countries and 75 in German speaking countries); mean age: 36 years (SD = 8 years); Unclear specific follow up time; Multiple cancer diagnoses, i.a. cervical cancer (45%), and breast cancer (30%)	A study specific web-based questionnaire	Knowledge about FP was limited among participants. Confidence of knowledge was significantly higher in women who had undergone any FP procedure. Greater emphasis should be placed on counselling opportunities and the provision of adequate information and support materials.
Van den Berg et al. [54]	The Netherlands	2022	To systematically assess the quality of integrated female oncofertility care by patient-reported measurement, to measure which determinants were associated with this quality of care, and to seek to develop tailored improvement strategies to improve the quality of integrated female oncofertility care.	Quantitative	Retrospective	2020–2021	Female AYA cancer patients 18 up to and including 40 years of age, diagnosed in 2016 or 2017 and received a (potentially) gonadotoxic treatment.	2016–2017	*N* = 121/344 (35%); Mean age: 34 years	Study specific questionnaire	Four determinants (patient age, strength of wish to conceive, time before cancer treatment, and type of health care provider) were found to be indicators on referral and shared decision making. Higher patient age was associated with lower referral rates. A higher wish to conceive was associated with higher referral rates and receiving written and/or digital information.
Vogt et al. [69]	UK	2018	To investigate factors influencing the decisions women with new diagnoses of cancer make about their fertility and to compare the quality of life, levels of anxiety, depression, illness perceptions, and optimism between women who chose to preserve their fertility and those who do not.	Mixed methods	Prospective	Unclear	Women (aged 16–40 years) with a new diagnosis of cancer and planned potentially gonadotoxic treatment (chemotherapy and/or radiotherapy). Group 1: from oncology who chose not to be referred to the assisted conception unit; Group 2: recruited from the ACU to see a fertility expert; Group 2A: those who made a positive FP decision; Group 2B: those who did not undergo FP.	Unclear	*N*: group 1 = 34; Mean age: 34 years; Multiple diagnoses, i.a. breast cancer (62%) and cervical cancer (21%); *N*: group 2 = 23; Mean age: 29 years; Multiple diagnoses, i.a. breast cancer (61%) and lymphoma (17%); Semi-structured interviews (group 2) *N* = 14/23 (61%); Mean age: 31 years; Multiple diagnoses, i.a. breast cancer (79%)	Questionnaires in two groups Group 1: Hospital anxiety and depression scale (HADS) and a short study-specific decision-making questionnaire (only ones) Group 2: 5 times during the car pathway to measure aspects of decision -making, patient satisfaction and HRQoL (validated questionnaires) and Semi-structured interviews to explore their experiences of the FP process.	Five themes were identified: Timing and quality of information provision Psychosocial factors Age Clinical influences Financial costs
Walasik [55]	Poland	2023	To assess the experience of Polish female cancer patients regarding FP after undergoing gonadotoxic treatment.	Quantitative	Retrospective	2020	Aged 18–50 years at the time of completing the survey, aged 10–40 at the time of malignancy diagnosis, diagnosis no more than 10 years before completing the questionnaire.	Unclear	*N* = 299; Multiple diagnoses, i.a. breast cancer (38%), Hodgkin lymphoma (22%) and thyroid cancer (9%)	Study specific questionnaire distributed via internet	More than half of the female subjects in this study did not undergo any pre-treatment FP counselling, although half of them had never been pregnant before the cancer diagnosis. Almost 30% claimed to have started the discussion. Around one third of the women had been referred to a fertility specialist, although only half of them visited the specialist. Half of the participants said that FP was not an important issue, instead seeing their oncological treatment as the highest priority. Women that were concerned about the negative influence of their cancer treatment on their fertility were more likely to use any form of FP. Only 17% of the participants did use some kind of FP. The study summarized reasons for not using FP and questions about the safety of having children after cancer treatment, as well as discussing the high cost of such treatment and the lack of knowledge thereof among some patients.
Von Wolff et al. [56]	Germany, Switzerland and Austria	2016	To assess patients’ attitudes about their fertility and about the counselling process at the time when FP counselling was performed	Quantitative	Cross-sectional	2012–2013	Female cancer patients aged 18–43 years, after FP counselling and before starting a gonadotoxic treatment. Patients were recruited in five centers belonging to the FertiPROTEKT network.	2012–2013	*N* = 144; Mean age: 30 years; Multiple diagnosis, i.a. breast cancer (54%) and lymphoma (22%)	Study specific questionnaire	As fertility concerns and attitudes about the counselling process were found to be independent of the chosen FP procedure, the preferred treatment can not accurately be predicted, and therefore, all women should be counselled about all possible FP techniques. Counselling by specialists about FP techniques is essential for all women undergoing gonadotoxic treatment, irrespective of whether they decide against or for a specific FP treatment.
Yee et al. [29]	Canada	2012	A survey of the views of female cancer survivors who sought an FP consultation prior to commencing cancer treatment.	Qualitative	Retrospective	Unclear	Female cancer patients referred to an IVF clinic for FP consultation between 2005 and 2008.	2005–2008	*N* = 41/70 (59%); Mean age: 33 years; Multiple diagnoses, i.a. breast cancer (76%), ovarian cancer (5%), and lymphoma (5%).	Study specific questionnaire The questionnaire also included five open-ended questions	It is important for oncology health care providers to initiate a discussion with all reproductive-age cancer patients. Timely referral to a fertility specialist is essential. It is beneficial to receive background information on FP options prior to the meeting with a fertility specialist. There is a lack of accessible and reliable cancer-related FP resources to reduce service barriers. Empathetic communication and flexibility in terms of accommodating fertility care needs are important.
Zanagnolo et al. [57]	Italy	2005	To evaluate the reproductive history, the experiences, attitudes, and emotions with regard to having children in conservatively treated patients with stage I epithelial ovarian cancer, any stage low malignant potential (LMP) tumors, malignant ovarian germ cell tumors (MOGCTs), or stage I sex cord-stromal tumors (SCSTs)	Quantitative	Retrospective	Unclear	Aged < 40 years at the time of diagnosis of stage I epithelial ovarian cancer, any stage low malignant potential (LMP) tumors where at least one of the ovaries was not or was only minimally involved, MOGCTs, or stage I SCSTs. Primary conservative surgical treatment, diagnosed between 1986 and 2000.	1986–2000	*N* = 41/68 returned questionnaires (60%); Mean age: 25 years at diagnosis; Median time of follow up: 102 months (35–192 months); Malignant ovarian tumors	Study specific questionnaire	Infertility has a strong potential to cause distress. Health care providers should be sensitized to the need to counsel patients proactively about reproductive issues.
Male											
Achille et al. [30]	Canada	2006	To explore factors that facilitate or hinder sperm banking among survivors of testicular cancer and Hodgkin’s disease.	Qualitative	Retrospective	Unclear	Male patients aged 2–10 years post-diagnosis, being adult (age ≥ 18 years) at the time of diagnosis, having received chemotherapy alone or as part of a combination treatment for either testicular cancer or Hodgkin’s disease.	Unclear	*N* = 20; Mean age: 32 years at interview (27 years at diagnosis); Diagnosis of testicular cancer or Hodgkin’s disease	Semi-structured interviews	Six factors were identified as having an impact on sperm banking: the role of health care providers in discussing infertility and sperm banking, the importance survivors place on having biological children (and fatherhood status at the time of diagnosis), the influence of a parent or partner, attitudes toward survival at the time of diagnosis, the cost of sperm banking, and perceptions about the complexity and efficacy of sperm banking.
Chapple et al. [31]	UK	2007	To examine the experiences and perceptions of young men who have had cancer and who now have had to cope with fertility issues.	Qualitative	Retrospective	2004–2005	Male patients diagnosed with cancer.	Unclear	*N* = 18, 6 between 16–18 years at interview and 12 between 19–26 years at interview; Multiple but unclear diagnoses.	Semi-structured interviews	Four themes appeared to be salient to the young men interviewed: the importance of choice, the need for more counselling, concerns about sperm banking, and feelings about possible infertility.
Edge et al. [58]	UK	2005	To identify what proportion of patients was able to store semen and which factors affected their success of failure.	Quantitaive	Retrospective	2001	Male patients aged 13–21 years at the time of diagnosis. Diagnosed between 1997 and 2001.	1997–2001	Response rate: *N* = 45/55 completed questionnaires (82%); Multiple diagnoses, i.a. Hodgkin’s disease (27%), osteosarcoma (22%) and testicular tumors (16%); Mean age at time of diagnosis: 17 years; Average interval between diagnosis and completing the questionnaire: 2 years	Study specific online and additional focus-groups raising additional topics in face-to face discussions	The majority of adolescent cancer patients are able to store viable semen if offered the opportunity. Those who failed to bank sperm were younger, had greater levels of anxiety at diagnosis, and had more difficulty talking about fertility. Semen cryopreservation should be offered as a routine procedure to all sexually mature adolescents that are at risk of fertility impairment.
Ehrbar [70]	Switzerland	2022	To examine how male cancer patients experienced counselling and how helpful they perceived it to be. Their counselling needs were assessed and evaluated and they were asked whether an online support tool could be helpful.	Mixed Methods	Retrospective	Unclear	Men above 18 years of age with a cancer diagnosis within the last 10 years and 13 years or older at the time of diagnosis.	Unclear	Response rate: *N* = 72/149 completed online questionnaire (48%) and 12 were part of the following 3 focus-groups; Mean age: 33 years (31 years at diagnosis); Multiple diagnoses, i.a. testicular cancer (56%), Lymphoma (17%), and leukemia (14%)	Study specific online questionnaire and additional focus groups raising additional topics in face-to-face discussions	Cancer patients undergoing gonadotoxic therapy should be counselled about FP. Male participants appreciated a well-organized FP process. An online support tool that provides information about FP and general reproductive health was considered to be helpful.
Krouwel et al. [59]	The Netherlands	2021	A survey regarding patients’ experiences of discussions of fertility concerns and sperm preservation, the procedure of sperm cryopreservation, the number of children and the use of preserved samples, and satisfaction levels regarding information provision and reproductive concerns.	Quantitative	Retrospective	2016	Patients who were or had been in treatment at an outpatient clinic of the urology and/or oncology department of a university hospital between 1995–2015, with pathologically confirmed testicular cancer in their medical history, aged 18–70 years.	1995–2015	Response rate: *N* = 201/582 (35%); Mean age: 44 years; Mean age at diagnosis: 34 years; Follow up time: 11 years; Single diagnosis testicular cancer	Study specific questionnaire	A majority of respondents were notified about the possibility of fertility problems as a result of their treatment (88%). However, the possibility of sperm cryopreservation was discussed with only 77% of respondents. The reported levels of satisfaction with care could be directly correlated to the amount of information provided regarding fertility risks.
Latif and al. [32]	Pakistan	2019	To identify patient- and physician-related factors that influence decision about sperm banking in cancer patients, with particular emphasis on cultural aspects.	Qualitative study	Retrospective	Unclear	Male cancer patients aged 18–45 years, irrespective of cancer stage or type; semi structured interviews.	Unclear	Response rate: *N* = 25/31 (81%); Mean age: 31 years; Multiple diagnoses, i.a. Leukemia 40% and lymphoma 12%	Semi structured interviews	There is a significant lack of awareness among male cancer patients regarding infertility following cancer treatment. It is imperative that physicians inform them of this and discuss treatment options, along with addressing potential barriers.
Pacey [60]	UK	2013	To identify medical, demographic, and psychological variables on diagnosis (T1) and 1 year post-diagnosis (T2) which differentiate between bankers and non-bankers, and to determine health related quality of life among a sample of young men where treatment posed a risk to their fertility.	Quantitative	Prospective	2008–2010	Male cancer patients aged 18–45 years, diagnosed with either testicular cancer of a hematological disorder, good prognoses and undergoing treatment with curative intent.	2008–2009	Response rate: *N* = 91/105 (87%) (T1), 78/91 (86%) (T2); Mean age: 33 years; Two diagnoses, i.e., testicular cancer and hematological disorder	Multiple questionnaires: Health-related quality of life (QLQ-C30), Princess Margaret Hospital Patient Satisfaction with Doctor Questionnaire (PMH/PSQ-MD), Brief Illness perception questionnaire-revised (BIPQR) and study specific questionnaire	Patients who underwent sperm banking were younger and less likely to have children than non-bankers. Extra care should be taken when counselling younger men who may have given little consideration to future parenting. The results support a previous finding, i.e., that the role of a health care provider is vital in facilitating decisions, especially for those who are undecided about whether they want children in the future or not.
Perez et al. [61]	Canada	2018	To describe and examine the fertility-related informational needs of male cancer patients, to describe FP practices, as well as perceived barriers and facilitators to sperm banking among male cancer patients, and to examine if demographic characteristics were significantly related to fertility discussions and FP practices.	Quantitative	Retrospective	2015–2016	Male cancer patients aged 18–55 years.	Unclear	Response rate: *N* = 192/274 completed questionnaires (70%); Mean age: 34 years; Average age at cancer diagnosis: 30 years; Multiple diagnoses, i.a. lymphoma (40%), sarcoma (14%), and testicular cancer (14%)	Study specific questionnaire	Misconceptions about passing on cancer to one’s child and that sperm cryopreservation will delay treatment should be dispelled. Health care providers can ask patients if they have any desire to have children in the future as way to initiate a discussion about FP. Key information gaps and psychosocial resource needs are suggested to fully meet male cancer patients’ fertility-related concerns.
Schover et al. [62]	USA	2002	To confirm and elaborate the findings of a pilot study, i.e., that only 19% of men had banked sperm and that the most common reason for not banking was a lack of information.	Quantitative	Retrospective	Unclear	New diagnosis of cancer, aged 14 to 40 years. Treatment with chemotherapy, radiation to the whole body, pelvis, brain, or abdomen, or having intent to undergo pelvic surgery.	Unclear	Response rate: *N* = 201/904 completed questionnaires (22%); Mean age: 30 years; Response on average 3 years after cancer diagnosis; Multiple diagnoses, i.a. leukemia (24%), lymphoma, (26%) and testicular cancer (11%)	Study specific questionnaire	This study confirmed the importance of fatherhood to younger male cancer survivors. Over 50% of men would like to have a child in the future, including three quarters of men who were childless at diagnosis.
Xi et al. [63]	China	2020	To assess the awareness of fertility protection among patients and healthcare providers.	Quantitative	Retrospective	Unclear	Male cancer patients, aged 15–45 years.	Unclear	Response rate: *N* = 407/500 patients completed and returned questionnaire (81%); age during treatment and age since cancer diagnosis (unclear); Cancer type (unclear)	Study specific questionnaires	The awareness of reproductive protection among both physicians and cancer patients in Qingdao, China, was not satisfactory. There is a need for comprehensive health education and practical protocols.
Yee et al. [64]	Canada	2012	To explore factors associated with oncology patients’ decision to bank sperm prior to cancer treatment.	Quantitative	Cross-sectional	2009–2010	Patients referred for an oncology sperm banking program in the 2009–2010 period.	2009–2010	Response rate: *N* = 79/157 patients (50%); Mean age: 28.4 years; Multiple diagnoses, i.a. testicular cancer (35%), Hodgkin’s lymphoma (14%), and non-Hodgkin’s lymphoma (13%)	Study specific questionnaire, with more choice questions and some open-ended questions.	Two key determinants are associated with the sperm banking decisions: the physician’s recommendations and the patient’s desire for future fatherhood.
Zhang et al. [65]	China	2020	To explore the FP-related knowledge and needs of male cancer patients of reproductive age.	Quantitative	Cross-sectional	2017–2018	Male patients aged 18–45 years, on initial admission to the hospital and undergoing or already finished treatments that threaten fertility.	2017–2018	*N* = 332 patients completed the questionnaire; Mean age: 35.5 years; Multiple diagnoses, i.a. colorectal cancer (45%), malignant lymphoma (27%), and prostate cancer (23%)	Study specific questionnaire.	Knowledge about FP in male cancer survivors of reproductive age is generally poor. During treatment, some patients were interested in obtaining more information regarding FP.
Female and male											
Anazodo et al. [33]	Australia	2016	To explore the experiences of consumers of oncofertility, to identify areas of oncofertility care needing development, and to develop a “charter” of the values and goals of consumers and health care providers regarding the gold standard in oncofertility care.	Qualitative	Retrospective	2014–2015	Pediatric, adolescent, and young adult cancer patients, completed treatment and in remission.	Unclear	*N* = 32, aged between 15–46 years, median age 22 years, 14 male, 18 female; Multiple diagnoses, i.a. Hodgkin’s lymphoma (22%), testicular cancer (16%), and acute lymphoblastic leukemia (16%).	Focus group discussion	Health care providers should discuss the possible effects of cancer treatment on a patient’s fertility before the start of treatment, irrespective of age, diagnosis, and prognosis of the patient. They should give patients an opportunity to discuss this by offering a referral to a fertility specialist. There should be a clear referral pathway to ensure that FP consultations can be organized in a timely manner. FP strategies should be affordable and equitable for all cancer patients. Psychosocial support should be available.
Armuand et al. [34]	Sweden	2014	To investigate newly diagnosed cancer patients’ experiences regarding fertility-related communication and their reasoning based on the risk of future infertility.	Qualitative	Cross-sectional	2009–2011	Patients aged 20–45 years, newly diagnosed with cancer (within a few weeks following diagnosis), planned treatment regarded as curative and with potential negative impact on fertility (2009–2011).	2009–2011	Response rate: *N* = 21/29 patients agreed to participate (72%); *N* = 11 women (median age 32 years) 10 men (median 33 year); Multiple diagnoses, i.a. lymphoma (24%), breast cancer (19%), and leukemia 19%)	Semi-structured interviews	Adequate information: very little, none, or only written fertility related information. Unmet informational needs about fertility preservation, as well as questions regarding the use of contraceptives during cancer treatment and how to check one’s fertility status after completed treatment.
Canzona et al. [35]	USA	2021	To construct a conceptual model of fertility concerns for AYAs.	Qualitative	Retrospective	Unclear	Patients diagnosed with cancer as an AYA, currently receiving treatment or within 5 years of completing treatment.	Unclear	*N* = 36 (Adolescents *N* = 10, mean age 17, Emerging adult *N* = 12, mean age 21, Young adult *N* = 14, mean age 33), 16 male, 20 female; Multiple diagnoses, i.a. leukemia (28%) and lymphoma (19%)	Semi-structured interviews	Four domains (affective, informational, coping, and logistical) of themes characterize fertility concerns among AYAs with cancer. AYA fertility and FP experiences were shaped by communication and timing factors. AYA fertility concerns are characterized by uncertainty and confusion that may contribute to future decisional regret or magnify feelings of loss.
Canzona et al. [36]	USA	2023	To explore ways in which AYAs with cancer may experience turning points throughout the FP decision making process, with a particular focus on differences between non-Hispanic White and racial/ethnic minority patients.	Qualitative	Retrospective	Unclear	Aged 15–39 years, currently receiving cancer treatment or within 5 years of concluding treatment.	Unclear	*N* = 36/68, 20 non-Hispanic White (11 men, 9 women, mean age 25 years), 16 racial/ethnic minority (5 men, 11 women, mean age 25%); Multiple diagnoses, i.a. leukemia (36%), Lymphoma (19%), and sarcoma (14%)	Semi-structured interviews (in person (*N* = 29) or via video/phone conference (*N* = 7)	Seven thematic turning points are described: (1) emotional reaction to discovering that FP procedures exist, (2) encountering unclear or dismissive communication during initial fertility conversations with health care providers, (3) encountering direct and supportive communication during initial fertility conversations with health care providers, (4) participating in critical family conversations about pursuing FP, (5) weighing personal desire for a child against other priorities/circumstances, (6) realizing that FP is not feasible, and (7) experiencing unanticipated changes in cancer diagnosis or treatment plans/procedures. Non-Hispanic white participants emphasized more forcefully that biological children may become a future priority.
Crawshaw et al. [37]	UK	2009	To investigate male and female adolescent cancer patients’ experiences in terms of fertility and associated decision-making matters being raised at diagnosis.	Qualitative	Retrospective	2004–2006	Diagnosed between 13–20 years or age, aware that fertility might have been affected and not receiving treatment.	Unclear	*N* = 38 (response rate of about 38%), 16 13–21 year-olds at interview (7 men, 9 women); median time since diagnosis: 3 years; 22 21–30 year-olds (12 women, 10 men); median time since diagnosis: 7 years); Multiple diagnoses, i.a. lymphoma (16%), leukemia (16%), and germ cell tumors (13%)	Semi-structured interviews	This study emphasizes the importance of addressing possible reproductive health implications at or around the time of diagnosis, even if options for fertility preservation are neither available nor appropriate, as well as the wish to have a choice in who should be included in these highly intimate discussions.
Kayiira et al. [66]	Uganda	2022	To establish the extent of self-reported reproductive failure associated with cancer treatment among AYA cancer survivors in Uganda and attitudes toward future fertility among AYA survivors of cancer.	Quantitative	Retrospective	Unclear	AYA survivors of cancer diagnosed between 2007 and 2018. At least 18 years of age, diagnosed with cancer between ages of 0 and 5 years.	2007–2018	*N* = 34, 14 females, 20 males; Median age at interview: 27 years (females) and 25 (males); Median age at diagnosis: 24 years (females) and 18 years (males); Multiple diagnoses, i.a. Kaposi’s sarcoma (44%), Burkitt lymphoma (9%), and Hodgkin’s lymphoma (6%)	Study specific telephonic interview questionnaire.	Information and counselling provided regarding therapy-related problems before cancer treatment was insufficient, reinforcing the need to build up the capacity for oncofertility resources within the region.
Levin et al. [38]	USA	2023	To explore the overall experiences of AYAs who encounter potential iatrogenic infertility and the ways in which financial concerns impact FP decision-making strategies.	Qualitative	Retrospective	2019–2020	Patients aged 12–25 years, diagnosed within the previous 2–12 months, at risk for infertility owing to diagnosis or prescribed curative treatment.	2018–2020	Response rate: *N* = 27/60 (45%), 17 males, 10 females; Multiple diagnoses, i.a. Hodgkin’s lymphoma (33%), synovial sarcoma (11%), and osteosarcoma (11%)	Semi-structured interview	Multiple and interrelated financial considerations factor into AYA experiences and decision making, including insurance coverage, presence of parental/guardian support, access to financial aid, negotiating potential risks, and consideration of long-term costs.
Marino et al. [67]	Australia	2023	To examine the perspectives of AYAs with cancer regarding the information they received about potential infertility and FP options and the decisions they made regarding their fertility.	Quantitative	Retrospective	2010–2012	6–24 months after a diagnosis of cancer (including first diagnosis, relapse, or diagnosis of second cancer).	2008–2012	*N* = 196 returned completed surveys, 99 males (51%), 97 females (49%); Mean age at diagnosis: 20 years; mean age at survey completion: 22 years; Multiple diagnoses, i.a. malignant hematological diseases (31%), Hodgkin lymphoma (25%), and sarcoma (15%).	Study specific questionnaire	Family involvement in decision-making was considered helpful. Older patients were more likely than younger ones to have involved partners, although AYAs will be the main decision makers with regard to FP, particularly as AYAs mature. Resources and support should be available for patients’ parents, partners, and siblings.
Parton et al. [39]	Australia	2019	How do women and men construct and experience FP treatment?	Qualitative	Retrospective	Unclear	People with cancer and their partners responded to advertisements regarding fertility care after cancer. (See also study Ussher et al. [71])	Unclear	Survey *N* = 693 women and 185 men; Average age: 43 years; average time from diagnosis: 6 years; Multiple diagnoses, i.a. breast cancer (57%), Gynecological cancer (13%), and hematologic cancer (13%); Interviews *N* = 61 women and 17 men; Mean age: 45 years	Survey with open answer questions and semi-structured telephone interviews	Three main discursive themes: limited agency and choice or resisting risk, FP as a means to retain hope and control, and FP as something that is uncertain and distressing.
Peddie et al. [9]	UK	2012	To explore perceptions about FP techniques among men and women of reproductive age, diagnosed with cancer in a tertiary referral center without the full range of facilities for cryopreservation.	Qualitative	Cross-sectional	2008–2010	Recently diagnosed men and women aged 16–44 years (2008–2010).	2008–2010	*N* = 16/18 men (89%), 18/21 women (86%); Mean age: 30 years; Multiple diagnoses, i.a. Hodgkin’s lymphoma (24%), leukemia (21%), and testicular cancer (15%)	Semi-structured interviews	Survival was always viewed as paramount, with future fertility being secondary. Few women were afforded the opportunity to discuss FP options, reflecting clinicians’ reservations about the experimental nature of egg and ovarian tissue cryopreservation and the need for partner involvement in embryo storage.
Salsman et al. [40]	USA	2021	To explore attitudes and practices about FP procedures.	Qualitative	Retrospective	Unclear	Patients aged between 18–39 years at diagnosis, within 2 years of treatment, and having met with a fertility navigator or reproductive specialist. Diagnosed with breast, gynecologic, neurologic, gastrointestinal, sarcoma, lymphoma, leukemia, or genitourinary/urologic cancer.	Unclear	Response rate: *N* = 24/49 patients (49%), 15 female, 9 male; Mean age: 29 years; Multiple diagnoses, i.a. leukemia (17%), lymphoma (17%), and brain (13%)	Semi-structured individual interviews	AYAs want to receive accurate and in-depth information. AYAs shared their experiences regarding the emotional impact of cancer-related infertility and the desire for support from trusted others.
Ussher et al. [71]	Australia	2018	Are there differences between women and men in terms of the degree of satisfaction with the communication of health professionals about fertility? How do men and women construct discussions about fertility with health professionals, and what are the reported consequences for subjective wellbeing? To examine the construction and subjective experience regarding communication with health professionals about fertility in the context of a cancer diagnosis/treatment.	Mixed methods	Retrospective	Unclear	People with cancer who responded to advertisements regarding fertility care after cancer. (See also study Parton et al. [39])	Unclear	(See study population, Parton et al. [39])	An online survey with questions with multiple choice answer format and open answer questions and semi-structured telephone interviews	Satisfaction with HCP communication was achieved when HCP was proactive in terms of informing participants about the possible consequences of cancer treatment on fertility, as well as being informative, clear, and accurate.
Wang et al. [41] Australia	Australia New Zealand	2020	To explore the oncofertility care experiences, reproductive concerns, and psychological health of newly diagnosed cancer patients, and to determine how access to oncofertility care may influence the emotional experience of potential infertility at this time.	Qualitative	Retrospective	2016–2018	Male and female cancer patients of reproductive age (15–44 years).	2015–2018	Response rate: *N* = 30/52 (58%), 70% female; Mean age: 27 years; Mean time since diagnosis: 5 months (range 2–10 months); Multiple diagnoses, i.a. breast cancer (17%), Hodgkin’s lymphoma (13%), and leukemia (13%)	Quantitative questionnaire and qualitative semi-structured interview	Five themes were identified: Satisfaction with oncofertility care Need for individualized treatment and support Desire for parenthood Fertility treatment can be challenging Fertility care provides a safety net for the future

AYA, adolescent and young adult; DCS, decisional conflict scale; FIS, Fertility Issues and outcomes scale; FP, fertility preservation; HADS, Hospital Anxiety and Depression Scale; OTC, ovarian tissue cryopreservation.

**Table 2 cancers-15-05828-t002:** Relevant PROs and patient experiences in FP.

PROs and Patient Experiences in Fertility Preservation
General Topics of Experiences and Needs in FP	Specific Topics of Experiences and Needs in FP among Female AYAs	Specific Topics of Experiences and Needs among Male AYAs
Starting a conversation about the risk of fertility decline and referral for FP counselling:-A health care provider should acknowledge the importance of future fertility;-Actively starting a conversation about potential fertility decline by health care provider;-(Early) Referral to a fertility specialist for FP counselling.		Health care providers should acknowledge that talking about and undergoing FP can be embarrassing.
The need for verbal and written (patient specific) information		
Psychosocial effects of facing potential fertility decline.	Effect of natural age related decline of fertility and FP.	
Experiencing FP-Counselling-Decisions about FP treatment Sense of controlHope and future orientationSource of distress	Feeling out of place in a fertility center;Risks specific to different cancer types.	
Worries about future fertility	Fear of missing the window of reproductive opportunity, fear of early menopause; Risks associated with ovarian tissue transplantation after cancer survival.	
Long term follow up after FP

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
