# Peer review of "Patient-Reported Outcomes (PROs) and Patient Experiences in Fertility Preservation: A Systematic Review of the Literature on Adolescents and Young Adults (AYAs) with Cancer"

_cancers, 2023, doi:10.3390/cancers15245828_

Round 1
Reviewer 1 Report (Previous Reviewer 1)
Comments and Suggestions for Authors
looks like the authors addressed all of my points.
Reviewer 2 Report (New Reviewer)
Comments and Suggestions for Authors
Highly important topic, patient reported outcomes, with focus on fertility preservation in a time where cancer directed therapies become more and more effective leading to improved OS. Congratulations on the idea and the excellent implementation!
This manuscript is a resubmission of an earlier submission. The following is a list of the peer review reports and author responses from that submission.
Round 1
Reviewer 1 Report
Comments and Suggestions for Authors
The authors present a systematic literature review about fertility counseling and preservation in young cancer patients. The authors claim to present PROMS and PREMS in AYA cancer, but I do not see these 2 aspects represented in the results.
I’m not sure if the authors confuse PROMs and PROs? They certainly do not report on PROMs or PREMs, which are standardized and validated tools to measure PROs. Such measures simply do not exist for fertility preservation in oncology. Moreover, the focus on AYA cancer is not justified or taken into account in the article search or selection (or at least not well described; see also below).
Mainly, I urge the authors to:
1. rephrase their scope toward ‘aspects of fertility counseling/preservation” (or something similar), but not PROMs or PREMs. Their review is a narrative summary of patients’ experiences surrounding fertility counseling and preservation on oncology.
2. rephrase the scope to cancer patients of reproductive age. I certainly agree that a focus on AYAs is logical, but the authors do not seem to have been stringent about AYA age, and therefore the title/ scope is misleading. I believe that many of the included studies have participants with a mean age between 15-39, but didn’t focus on AYAs in particular. How did the authors determine eligibility if studies included a broader age range? (see also comment about the methods)
Moreover, the results are written very assertive/ as if representing the “truth” (vs. studies reporting/ indicating/suggesting etc.). Sometimes, the results read more like a discussion or the authors’ personal opinion. An approach closer to what studies’ results section reported might be indicated.
Other comments:
Methods:
Like I said above, the AYA criterion is not explained (e.g., did you in/exclude studies if they included any participants younger than 18 or older than 39? Did the mean age count?
Please specify that in the Methods and Limitations accordingly.
It seems like some included studies had participants that were older at diagnosis (e.g., mean age at diagnosis around 35 likely has people above age 40 years at diagnosis).
Related to that: How did you determine eligibility of studies about survivors of childhood cancer? They likely included many people diagnosed between 15-18 years of age, but the authors seem to have excluded all studies with childhood cancer survivors, correct? What about studies focusing on those diagnosed during adolescence?
Given that age was not really considered in the selection and actual reporting of results, this review seems to focus on any patient of reproductive age, or not?
Search strategy:
“sperm freezing” is not typically used. And why did the authors not include “sperm banking”? I’m afraid the authors missed some references this way.
-and why did the authors not use the term ‘hematology’ ?
Accordingly, I think the authors may have missed studies from Canzona, Nahata, and some Scandinavian studies possibly too.
Quality assessment:
How did the authors determine the appropriateness of the research questions if a study was not specifically about fertility counseling, preservation, and/or AYAs?
The authors included studies published since 2002, but it appears that the authors did not consider year of study/ year of diagnosis, given that available FP options for female patients changed drastically over time. They only mention this in the Discussion section, but I would expect more in-depth information as part of the Results. Thereby, I would also suggest not considering the year of publication (as mentioned in the Discussion), but the year of diagnosis of included patients/ survivors or year(s) of recruitment.
Results:
All results are unsurprising, but I was surprised not to read about the following issue: Women can experience burden due to possibly having a shorter reproductive window and getting menopause earlier. This puts pressure on them to have children earlier. Again, this makes me question whether the authors missed certain studies?
- Some parts towards the end of the results feel repetitive
Discussion:
The authors mention the variety of included participants in the studies. Efforts to see whether similar groups reported similar outcomes/ experiences would have been interesting and adding to the depth of this review.
Comments on the Quality of English LanguageNext to editing the tone of the results section, I also urge the authors to check their grammar throughout the manuscript. There are many (although sometimes smaller) grammar issues, that make it sometimes hard to follow. Some examples:
Line 61: “are ranged” ? and what is the purpose of this whole sentence?
178: “in risk for infertility”
389: “proceed in”
392: “A few themes emerged in experiences in FP treatment.” What does this mean ?
495: “Patients have to put their fertility in trust to providers”
Author Response
Response to reviewer 1 comments
We thank the reviewer for his/her detailed review and legitimate comments on our manuscript. Concerns and questions have been addressed point-by-point below.
Point 1: The authors present a systematic literature review about fertility counseling and preservation in young cancer patients. The authors claim to present PROMS and PREMS in AYA cancer, but I do not see these 2 aspects represented in the results. I’m not sure if the authors confuse PROMs and PROs? They certainly do not report on PROMs or PREMs, which are standardized and validated tools to measure PROs. Such measures simply do not exist for fertility preservation in oncology. Moreover, the focus on AYA cancer is not justified or taken into account in the article search or selection (or at least not well described; see also below).
Mainly, I urge the authors to: Rephrase their scope toward ‘aspects of fertility counseling/preservation” (or something similar), but not PROMs or PREMs. Their review is a narrative summary of patients’ experiences surrounding fertility counseling and preservation on oncology.
Response 1. It is correct that we made a misconception in our article about PROMs and PREMs instead of PROs and patients’ experiences. Based on your comments we rephrased the scope of the review towards: “Patient related outcomes (PROs) and patients’ experiences in fertility preservation; systematic review of literature of Adolescents and Young Adults (AYAs) with cancer” and revised the complete manuscript for use of incorrect terminology.
Point 2: Rephrase the scope to cancer patients of reproductive age. I certainly agree that a focus on AYAs is logical, but the authors do not seem to have been stringent about AYA age, and therefore the title/ scope is misleading. I believe that many of the included studies have participants with a mean age between 15-39, but didn’t focus on AYAs in particular. How did the authors determine eligibility if studies included a broader age range? (see also comment about the methods)
Response 2. Based on your comments we clarified the inclusion criteria of in- and excluded studies in the methods section towards: “Inclusion criteria were original research articles in English language addressing PROs and patients’ experiences of AYAs regarding counseling, treatment and future fertility in FP. Articles with AYAs as defined by the authors are included as well as articles with patients between 15-39 years of age at diagnosis (or > 75% included patients in this age range). Exclusion criteria were studies that focused on awareness of FP options or referral pathways as well as articles about younger children (defined as < 15 years), articles about children and their parents, articles with main group patients older than 40 years of age as research population, and review papers”.
Point 3: Moreover, the results are written very assertive/ as if representing the “truth” (vs. studies reporting/ indicating/suggesting etc.). Sometimes, the results read more like a discussion or the authors’ personal opinion. An approach closer to what studies’ results section reported might be indicated.
Response 3: Based on this comment we revised the result section to an approach closer to what studies’ results section reported be indicated. We also checked the tone throughout the complete manuscript.
Point 4: Methods:
Like I said above, the AYA criterion is not explained (e.g., did you in/exclude studies if they included any participants younger than 18 or older than 39? Did the mean age count?
Please specify that in the Methods and Limitations accordingly.
It seems like some included studies had participants that were older at diagnosis (e.g., mean age at diagnosis around 35 likely has people above age 40 years at diagnosis).
Related to that: How did you determine eligibility of studies about survivors of childhood cancer? They likely included many people diagnosed between 15-18 years of age, but the authors seem to have excluded all studies with childhood cancer survivors, correct? What about studies focusing on those diagnosed during adolescence?
Given that age was not really considered in the selection and actual reporting of results, this review seems to focus on any patient of reproductive age, or not?
Response 4: For clarity we added more information about the inclusion based on patients’ age in method section of the manuscript. The reviewer is correct that we excluded all studies with young childhood cancer survivors.
Point 5: Search strategy:
“sperm freezing” is not typically used. And why did the authors not include “sperm banking”? I’m afraid the authors missed some references this way.
-and why did the authors not use the term ‘hematology’ ?
Accordingly, I think the authors may have missed studies from Canzona, Nahata, and some Scandinavian studies possibly too.
Response 5: Based on this comment we performed a re-run of the search strategy and added the term “sperm banking”. This did not result in new eligible articles. Also adding the term ‘hemotology’ did not result in extra eligible articles to include in this review. We checked our search for studies with authors Nahata and Canzona. The first author Nahata was found in our search, but studies with this author were excluded based on eligibility criteria. However, we couldn’t find articles of author Canzona in our search. We found that one article about this topic was published after our conducted search of March 2nd 2023 and for that reason not included. But another article of this author seemed to be eligible to include. With this extra information we took a closer look why we missed this article in our search. The term “communication” seemed to be the key word. Adding this term in our search resulted in extra hits. We screened these articles first on title and abstract and 14 articles were selected for full review. This resulted in 7 extra included studies in this review. We thank the reviewer for this valuable addition.
Point 6: Quality assessment:
How did the authors determine the appropriateness of the research questions if a study was not specifically about fertility counseling, preservation, and/or AYAs?
Response 6: All the included studies showed in their aims of the study the topic fertility preservation and for that they were determined appropriate for quality assessment.
Point 7: The authors included studies published since 2002, but it appears that the authors did not consider year of study/ year of diagnosis, given that available FP options for female patients changed drastically over time. They only mention this in the Discussion section, but I would expect more in-depth information as part of the Results. Thereby, I would also suggest not considering the year of publication (as mentioned in the Discussion), but the year of diagnosis of included patients/ survivors or year(s) of recruitment.
Response 7: Year of diagnosis was mentioned in the table of our first draft. However, this comments informs that this has not been clear enough. We changed the layout of the table and added a separate column ‘year of FP counselling’ to highlight this topic more clearly and also changed it in the manuscript.
Point 8: Results:
All results are unsurprising, but I was surprised not to read about the following issue: Women can experience burden due to possibly having a shorter reproductive window and getting menopause earlier. This puts pressure on them to have children earlier. Again, this makes me question whether the authors missed certain studies?
Response 8: With inclusion of the new articles this topic was more clearly documented and included in the revised version review.
Point 9: Some parts towards the end of the results feel repetitive
Response 9: We combined phrases, rearranged the flow of information and tried not to be repetitive in the information.
Point 10: Discussion:
The authors mention the variety of included participants in the studies. Efforts to see whether similar groups reported similar outcomes/ experiences would have been interesting and adding to the depth of this review.
Response 10: The group of FP patients is or course a very heterogenic group of patients. In line with your suggestion we tried to group the reported outcomes in general topics and male and female topics. In some groups (e.g. female ovarian tissue cryopreservation patients) more specific outcomes/experiences were reported. These topics were mentioned in our manuscript.
Point 11: Comments on the Quality of English Language
Next to editing the tone of the results section, I also urge the authors to check their grammar throughout the manuscript. There are many (although sometimes smaller) grammar issues, that make it sometimes hard to follow. Some examples:
Line 61: “are ranged” ? and what is the purpose of this whole sentence?
178: “in risk for infertility”
389: “proceed in”
392: “A few themes emerged in experiences in FP treatment.” What does this mean ?
495: “Patients have to put their fertility in trust to providers”
Reponse 11: The specific points noted in this comment were changed in the manuscript. Furthermore the grammar was checked throughout the complete manuscript.
We hope that all the made adjustments contribute to your agreement to a positive review report for our manuscript.
Yours sincerely,
N.F. Klijn
Reviewer 2 Report
Comments and Suggestions for Authors
In this Review the authors are trying to talk about PROMs and PREMs in fertility preservation esp in adolescents and young adults with cancer. The points raised by the authors are valid but they are not unique enough to be reported as a review. Its not something which is not known or may not be practiced. The authors themselves acknowledge in discussion that referral patterns are probably changed. This may be more suitable as a opinion piece but not as a review article.
Author Response
Response to reviewer 2 comments
We thank the reviewer for his/her review and comments on our manuscript. Concerns and questions have been addressed point-by-point below.
Point 1: In this Review the authors are trying to talk about PROMs and PREMs in fertility preservation esp in adolescents and young adults with cancer. The points raised by the authors are valid but they are not unique enough to be reported as a review. Its not something which is not known or may not be practiced. The authors themselves acknowledge in discussion that referral patterns are probably changed. This may be more suitable as a opinion piece but not as a review article.
Response 1: We understand your comment on the point raised and topic of our review. Still, we think that a systematic review on this topic helps to get a complete overview of all patient reported outcomes and patients' experiences in FP counselling, treatment and follow up. This overview can be used to create PROMs and PREMs that can be incorporated in a healthcare pathway for these patients to optimize care.
Yours sincerely,
N.F. Klijn
Reviewer 3 Report
Comments and Suggestions for Authors
Thank you for the opportunity to review this article which is a review of patient related outcome measures and patient reported experience measures in fertility preservation.
1. Introduction:
a. “In 2006 the term oncofertility was introduced by Teresa Woodruff to enlighten the importance of future fertility in oncological treatments”: authors should consider altering to “Dr. Teresa Woodruff to highlight the importance of discussing future fertility in oncological treatments.”
b. I also do not agree with the authors as to the meaning of the term “oncofertility”àthis term was coined to describe a field of medicine/research which described the interaction between cancer and fertility, and used to highlight the importance of studying and discussing potential fertility related outcomes with cancer treatments.
2. Line 178: should be “at risk for infertility”
3. Line 194: consider “starting a conversation or starting the conversation”
4. Line 201: change “initiating” to “initiation”
5. Line 218: change to “referrals should be sent”
6. Line 349: change “were” to “where”
7. Line 364-365: what do the authors mean by “reduced decision making”
8. Line 435: please specify what type of post cancer information the patients should be given/provided with
9. Line 522-523: “This could lead to a narrow view of all needs in patients specific needs in FP counselling and FP treatment”: please reword for clarity
10. Line 525: should say options, not optionLine 526-528: this is an incomplete sentence
11. Line 535-538: Peddie et al. (2012) for example described that women were feeling negative about fertility preservation because they were unsuitable for cryopreservation of gametes based on less possibilities in fertility preservation” please reword for clarity
12. Line 570-571: “This pathway should optimize patient experiences around FP and meet in the needed follow up after FP treatment”: please reword for clarity
Comments on the Quality of English Language
there are some grammatical/language errors throughout the manuscript, some of which I stated above. However, in general, this paper warrants a careful proofreading
Author Response
Response to reviewer 3 comments
We thank the reviewer for his/her detailed review and comments on our manuscript. Concerns and questions have been addressed point-by-point below.
Point 1: Introduction:
- “In 2006 the term oncofertility was introduced by Teresa Woodruff to enlighten the importance of future fertility in oncological treatments”: authors should consider altering to “Dr. Teresa Woodruff to highlight the importance of discussing future fertility in oncological treatments.”
- I also do not agree with the authors as to the meaning of the term “oncofertility”àthis term was coined to describe a field of medicine/research which described the interaction between cancer and fertility, and used to highlight the importance of studying and discussing potential fertility related outcomes with cancer treatments.
Response 1: We revised the introduction of our manuscript based on your comments and added extra reference about the oncofertility term: “In 2006 the term oncofertility was introduced by Dr. Teresa Woodruff to highlight the importance of discussing future fertility in oncological treatments. This term was coined to describe the intersection of oncology and fertility to help young women with cancer to protect their future reproductive health. The resulting oncofertility consortium is to provide fertility preservation options in the cancer setting”.
Point 2: Line 178: should be “at risk for infertility”
Response: Grammatical/language error has been changed
Point 3: Line 194: consider “starting a conversation or starting the conversation”
Response: The phrase has been changed in “Starting a conversation about potential fertility decline and referral for FP counselling”
Point 4: Line 201: change “initiating” to “initiation”
Response 4: Grammatical/language error has been changed
Point 5: Line 218: change to “referrals should be sent”
Response 5: Grammatical/language error has been changed
Point 6: Line 349: change “were” to “where”
Response 6: Grammatical/language error has been changed
Point 7: Line 364-365: what do the authors mean by “reduced decision making”
Respons 7: This phrase has been changed for clarity in : “The emotional and physical burden of cancer sometimes resulted in having reduced capacity for decision making and with that in uptake of FP”
Point 8: Line 435: please specify what type of post cancer information the patients should be given/provided with.
Response 8: We added information about the information patients should be provided with based on this comment: “Bach et al described that in a crisis and information overload at the point of diagnosis patients reported limited recollection and understanding of information received at the initial counselling[13]. Also in the study of Ehrbar et al. patients stated to be felt overwhelmed by the immense amount of information. The majority mentioned that it would be helpful to know that reproductive health they can be revisited later[62]. Patients in the study of Yee et al. also indicates that follow up after FP was important. It provided in depth information about sperm quality and better understanding of the results prior to the start of oncological treatment[59]”.
Point 9: Line 522-523: “This could lead to a narrow view of all needs in patients specific needs in FP counselling and FP treatment”: please reword for clarity
Response 9: This phrase has been changed for clarity in “This could possibly lead to a narrow view and with this missing of other patients’ specific needs in FP counselling and FP treatment”
Point 10: Line 525: should say options, not optionLine 526-528: this is an incomplete sentence
Response 10: Grammatical/language error has been changed
Point 11: Line 535-538: Peddie et al. (2012) for example described that women were feeling negative about fertility preservation because they were unsuitable for cryopreservation of gametes based on less possibilities in fertility preservation” please reword for clarity.
Respons 11: This phrase has been changed for clarity in : “Peddie et al. (2012) for example described that women were feeling negative about FP because they didn’t had the opportunity based on less FP possibilities at that time”
Point 12: Line 570-571: “This pathway should optimize patient experiences around FP and meet in the needed follow up after FP treatment”: please reword for clarity
Response 12: This phrase has been changed for clarity in “This patient centered approach will optimize experiences around FP and creates long term follow up after FP treatment”.
Point 13: Comments on the Quality of English Language
there are some grammatical/language errors throughout the manuscript, some of which I stated above. However, in general, this paper warrants a careful proofreading
Response 13: Thank you very much for your comments on the grammatical/language errors. The manuscript has revised and careful proofreading performed.
We hope that all the made adjustments contribute to your agreement to a positive review report for our manuscript.
Yours sincerely,
N.F. Klijn
Reviewer 4 Report
Comments and Suggestions for Authors
This article provides an extensive analysis of the current literature on the subject and is very weel written.
It may therefore be published in the present form
Author Response
Response to reviewer 4 comments
We thank the reviewer for his/her review and positive result on signing the review report.
Based on comments of other reviewers we revised our manuscript.
We hope that you also agree with the made adjustments for a positive review report for our manuscript.
Yours sincerely,
N.F. Klijn
Reviewer 5 Report
Comments and Suggestions for Authors
I think the systematic review performed is very good. I only wish more emphasis on the difficult to conclude due to the narrative nature of the review and the difficulty in getting more robust results.
Comments on the Quality of English Language
Good English
Author Response
Response to reviewer 5 comments
We thank the reviewer for his/her review and comments on our manuscript. Concerns and questions have been addressed point-by-point below.
Point 1: I think the systematic review performed is very good. I only wish more emphasis on the difficult to conclude due to the narrative nature of the review and the difficulty in getting more robust results.
Response 1: We do agree that the narrative nature of our review makes it difficult to give hard conclusions, however it summarizes the important patient related outcomes and experiences in AYAs which can help to optimize health care pathways for these patients.
Based on comments of other review we revised out manuscript.
We hope that you also agree with the made adjustments for a positive review report for our manuscript.
Yours sincerely,
N.F. Klijn
Round 2
Reviewer 1 Report
Comments and Suggestions for Authors
I do appreciate the authors’ efforts to address my comments. However, I still have several concerns:
1. There is still an inconsistency of how the authors refer to PROs. The title reads “patient related” outcomes, the text “patient reported” outcomes
2. Line 152 and Table 2 still claims that this review presents PROMs and PREMs.
3. The search: I appreciate the authors’ efforts to check for missing search terms, and now the manuscript reads: “A re-run of the search was performed.” This sounds like you updated your search for articles published after march 2023, which is not what you did. This whole process should be described differently in the paper.
4. Inclusion: The authors are clearer about their in/exclusion regarding age at diagnosis, but this part is confusing: “Articles with AYAs as defined by the authors are included as well as articles with patients between 15-39 years of age at diagnosis” Why are you adding these two parts? –your definition is also 15-39 years, right? What is the purpose of this sentence ?
I also can’t shake the feeling that you’re still missing studies if you used age 15-39. For example, studies focusing only on adolescents would also fall into this age category.
5. Results, Line 302: I was intrigued by the name “Zalagnolo et al” which was added to the results section, but this author is not mentioned anywhere else in the manuscript. In fact, if I google this name, I don’t find anything. How can such errors occur?
Overall, I do appreciate the authors’ efforts to update their search and results, but such issues are worrisome. It is also hard to be confident in the thoroughness of your search and results if 7 additional studies have been found (how much more did you miss?).
- Also, “Wang et al” is not listed in the reference list
6. How FP options changed over time for women is still not discussed.
7. Few edits have been made in the Discussion section. I don’t see much of an effort to address my previous point and discuss whether certain findings below to certain subgroups
Comments on the Quality of English Language
improved, still shaky at some points
Reviewer 2 Report
Comments and Suggestions for Authors
The reviewer acknowledges the author's attempts to include more data for analysis. However, the reviewer still believes that there is nothing novel about this to be published as a review article but can be an opinion piece.
Comments on the Quality of English LanguageMany sentences in the manuscript are difficult to understand because of grammatical errors/wrong sentence formation.